# Altered expression of maize *PLASTOCHRON1* enhances biomass and seed yield by extending cell division duration

Xiaohuan Sun[1,2], James Cahill[3], Tom Van Hautegem[1,2], Kim Feys[1,2], Clinton Whipple[1,2,4], Ondrej Novák[5,6], Sofie Delbare[1,2], Charlot Versteele[1,2], Kirin Demuynck[1,2], Jolien De Block[1,2], Veronique Storme[1,2], Hannes Claeys[1,2], Mieke Van Lijsebettens[1,2], Griet Coussens[1,2], Karin Ljung[7], Alex De Vliegher[8], Michael Muszynski[3], Dirk Inzé[1,2,*] & Hilde Nelissen[1,2,*]

Maize is the highest yielding cereal crop grown worldwide for grain or silage. Here, we show that modulating the expression of the maize *PLASTOCHRON1* (*ZmPLA1*) gene, encoding a cytochrome P450 (CYP78A1), results in increased organ growth, seedling vigour, stover biomass and seed yield. The engineered trait is robust as it improves yield in an inbred as well as in a panel of hybrids, at several locations and over multiple seasons in the field. Transcriptome studies, hormone measurements and the expression of the auxin responsive DR5rev:mRFPer marker suggest that PLA1 may function through an increase in auxin. Detailed analysis of growth over time demonstrates that PLA1 stimulates the duration of leaf elongation by maintaining dividing cells in a proliferative, undifferentiated state for a longer period of time. The prolonged duration of growth also compensates for growth rate reduction caused by abiotic stresses.

[1] Department of Plant Systems Biology, VIB, 9052 Gent, Belgium. [2] Department of Plant Biotechnology and Bioinformatics, Ghent University, 9052 Gent, Belgium. [3] Department of Genetics, Development, and Cell Biology, Iowa State University, Ames, 50011 Iowa, USA. [4] Department of Biology, Brigham Young University, Provo 84602, Utah, USA. [5] Laboratory of Growth Regulators and Department of Chemical Biology and Genetics, Centre of the Region Haná for Biotechnological and Agricultural Research, Palacký University, 771 47 Olomouc, Czech Republic. [6] Institute of Experimental Botany, Academy of Sciences of the Czech Republic, 771 47 Olomouc, Czech Republic. [7] Umeå Plant Science Centre, Department of Forest Genetics and Plant Physiology, Swedish University of Agricultural Sciences, 907 36 Umeå, Sweden. [8] Institute for Agricultural and Fisheries Research (ILVO), 9820 Merelbeke, Belgium. * These authors contributed equally to this work. Correspondence and requests for materials should be addressed to D.I. (email: diinz@psb.vib-ugent.be).

Worldwide, more metric tons of maize are produced annually than any other cereal crop and this is largely due to improvements in stover biomass and seed yield. These improvements were driven by modulation of several molecular mechanisms including transcriptional regulation[1], photosynthesis[2], hormone signalling[3] and carbon metabolism[4]. We explored if modulating the primary growth control processes of cell division or cell expansion might be an alternative approach to improve crop yield. The majority of the known growth control mechanisms are regulated in a cell-autonomous manner[5,6], including the proteolytic control of cell cycle regulators[7], expansins during cell expansion[8] and meristemoid cell divisions through PEAPOD (ref. 9). In addition to cell-autonomous regulators, phytohormones are well documented non-cell autonomous growth control determinants[10], and transcriptional activators, such as ANGUSTIFOLIA3, were also shown to control growth over a longer distance[11]. Our previous studies, using kinematic analysis of the maize leaf, which quantified the relative contribution of cell division and cell expansion during steady state growth[12], identified several maize growth regulators[13–15]. Of these, gibberellins (GAs) were shown to play an important role, because lines overexpressing the rate-limiting GA biosynthetic enzyme GA20-OXIDASE had larger leaves due to an increase in the leaf elongation rate (LER) and the number of dividing cells[13]. More recently, a cytochrome P450 CYP78A, named KLUH, was shown to regulate vegetative and reproductive organ growth across plant species by promoting cell proliferation likely through generation of a mobile growth-promoting signal[16–21]. In rice, PLASTOCHRON1 (PLA1) that belongs to the same class of CYP78A as KLUH affects the timing of leaf initiation and vegetative growth[22].

Here, we demonstrate ZmPLA1 is a GA-independent time-keeper of cell division. Constitutive overexpression of ZmPLA1 severely affected plant architecture by generating very large leaves and preventing reproduction. More localized ectopic expression of ZmPLA1 resulted in fertile maize plants that showed an increase in growth, stover biomass accumulation and seed yield. Using the maize leaf as a reporter, we demonstrated that localized ectopic expression of ZmPLA1 resulted in an increased duration of the maximal growth rate during steady state growth. The PLA1-mediated growth promotion also buffered against growth repression caused by two distinct abiotic stress conditions. Transcriptome studies, hormone measurements and the DR5rev:mRFP marker line suggest auxin contributes to this compensatory growth mechanism.

## Results

**Overexpression of PLA1 stimulates dramatic leaf growth.** The two well-characterized members of the CYTOCHROME P450 78A family that were described to date, KLUH and PLA1, affect plastochron and vegetative growth[22–24]. We identified a maize member of the CYP78A family, orthologous to the rice PLA1 (Supplementary Fig. 1), based on a very specific expression profiling along the maize leaf growth zone. In steady state growing leaves, PLA1 is most strongly expressed at the base of the leaf, particularly at the very base of the division zone (first half cm) (Supplementary Fig. 2a). Two independent transgenic maize events that constitutively overexpressed ZmPLA1 under control of the UBIL promoter (UBIL:PLA1-P1 and UBIL:PLA1-P2; Supplementary Fig. 3a) grew slower than wild-type B104 and failed to flower, preventing phenotypic analysis of progeny (Fig. 1). Therefore, phenotypic analyses were performed on the primary transformants and compared with B104 plants.

The UBIL:PLA1 events produced long and broad leaves (Supplementary Fig. 3a–c) that resulted in a pronounced increase in blade area (Fig. 1a,b). This dramatic difference in leaf size severely affected the plants' morphology and stature (Fig. 1c).

The mature cell length of leaf four of both transgenic events was significantly ($P < 0.001$, Student's $t$-test, $n > 340$) reduced compared with that of B104 (Fig. 1d), suggesting that constitutive expression of PLA1 stimulated cell division, producing more, but smaller cells.

**Localized PLA1 expression increases biomass and seed yield.** To expand the very narrow expression domain of PLA1 along the growth zone and to obtain a more subtle PLA1 expression than obtained by constitutive overexpression, we cloned the PLA1 gene behind the 2,046-bp promoter sequence of the GA2-oxidase (ZmGA2ox) gene that was shown to be enriched at the transition from cell division to cell expansion (Supplementary Fig. 2a)[13].

Three independent, single-locus events showed ectopic overexpression of PLA1 in the growth zone (GA2ox:PLA1-P1, GA2ox:PLA1-P2, GA2ox:PLA1-P3), of which GA2ox:PLA1-P3 had the highest expression level (Supplementary Fig. 2b). All three independent GA2ox:PLA1 transgenic maize plants were fertile and had no striking morphological abnormalities. However, these plants were taller (Figs 2b and 3a–c,e) and produced leaves with successively increased area, length and width (Fig. 2a,c–e), that were more pronounced with increasing leaf number (Fig. 2f,g). The transgenic lines were delayed in flowering, which was more pronounced for silking (72 days in GA2ox:PLA1 and 66 days in non-transgenic plants) than for pollen shed (74 days in GA2ox:PLA1 and 71 days in non-transgenic plants), resulting in a shorter anthesis-silking interval (on average 1.4 days in GA2ox:PLA1 versus 4.7 days in non-transgenic siblings; $P < 0.01$, Student's $t$-test, $n = 3$) (Fig. 3a). The enhanced leaf phenotypes persisted in hybrids originating from crosses between homozygous GA2ox:PLA1-P2 transgenic (T) to the inbred lines CML91, H99, F7, Mo17 and W153R but not from crosses of non-transgenic (NT) siblings to these same inbreds (Supplementary Table 1).

The presence of the GA2ox:PLA1 construct had a positive effect on stover biomass, seed yield and the synchronization of flowering in the B104 inbred as well as the B104xCML91 hybrid background in field conditions at two distinct locations over multiple growing seasons (Fig. 3, Supplementary Fig. 4). The leaf phenotypes were confirmed in field trials with the transgene in B104 inbred (US, 2014 and 2015; Supplementary Fig. 4) and in a hybrid background originating from crosses between transgenic (and non-transgenic) plants in B104 background with CML91 (USA, 2015; Belgium, 2015) (Fig. 3b–f; Supplementary Fig. 4). Plant fresh weight increased significantly in the transgenic hybrids compared with their non-transgenic hybrids (8–35%; Fig. 3c) due to increases in leaf length, width and blade area (measured for leaf four and ear leaf), plant height and stem width (Fig. 3c–f). In addition, moderate but consistent increases in the number of kernels per row (7–14%), the volume of the individual kernels (6–16%), ear length (17–26%) and cob dry weight (18–27%) were observed, relative to the non-transgenic controls in all field evaluations (Fig. 3b,d; Supplementary Figs 5,6a–c). The number of cobs per plant did not differ significantly between the transgenic and non-transgenic plants in the Iowa field trial ($P$ value $= 0.078$, Student's $t$-test, $n > 26$) and for GA2ox:PLA1_P1 in Belgium ($P$ value $= 0.064$, Student's $t$-test, $n > 120$ per plot), but the GA2ox:PLA1_P2 transgenic plants in the Belgium field trial produced significantly more cobs compared to the non-transgenic plants ($P$ value $\leq 0.05$, Student's $t$-test, $n > 120$ per plot; Supplementary Fig. 7).

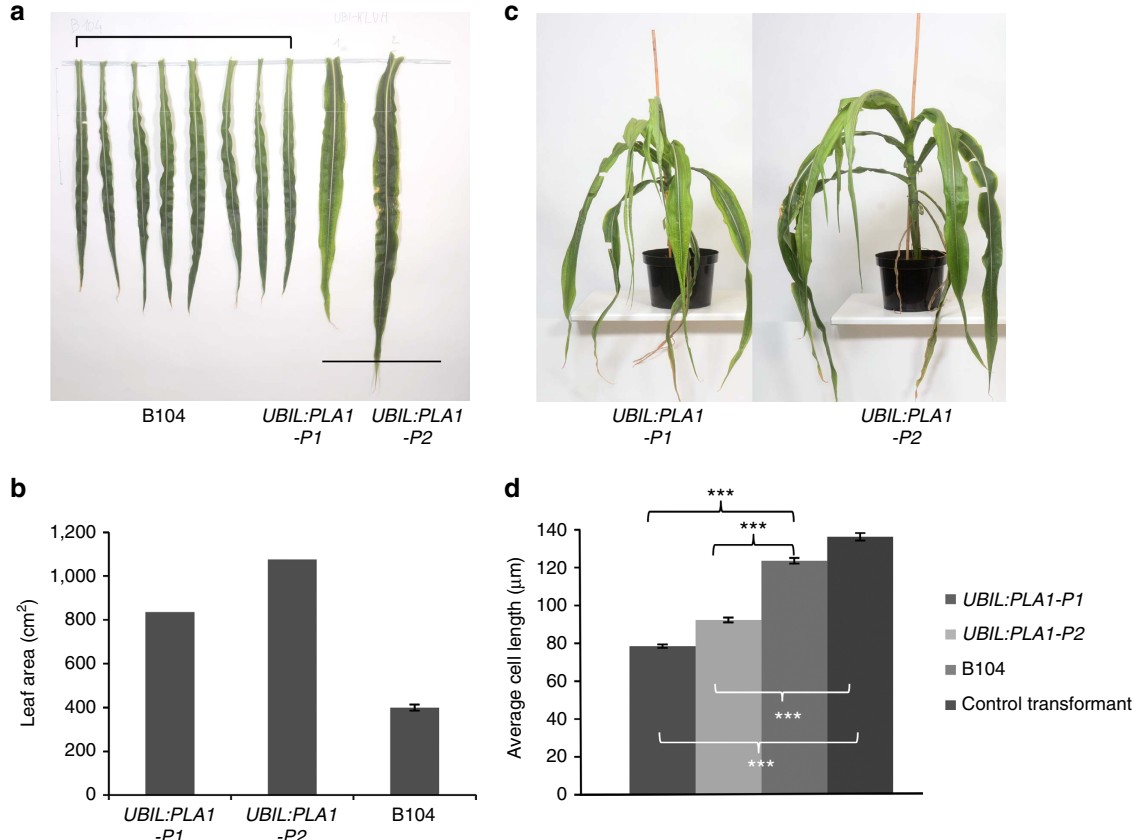

**Figure 1 | Phenotypes of the *UBIL:PLA1* plants.** (**a**) Fully grown ninth leaf blade from eight B104 plants compared with *UBIL:PLA1-P1* and *UBIL:PLA1-P2*. Scale bar indicates 50 cm. (**b**) Leaf blade area of the ninth leaf of the two *UBIL:PLA1* plants relative to the average of ten B104 plants. (**c**) *UBIL:PLA1-P1* at 121 days and *UBIL:PLA1-P2* at 135 days after transfer of the shoots to soil (compared with a normal life cycle of 70 days). (**d**) Average epidermal mature cell length of *UBIL:PLA1* and *UBIL:PLA1-P2* plants compared with the B104 control plants. Error bars indicate standard errors. Significant differences ($P < 0.001$, Student's $t$-test, $n > 8$) are indicated with asterisks (\*\*\*).

**PLA1 stimulates growth by extending cell division duration**. Typically, the effects of perturbations affecting growth are quantified by performing a kinematic analysis during steady-state growth of the maize leaf[12]. Strikingly, kinematic analysis of *GA2ox:PLA1-P1* and *GA2ox:PLA1-P2* revealed no statistically significant differences in cell production, the size or number of dividing cells, or the size of the division zone (Supplementary Table 2) and the effect on mature cell size was too small to explain the 19–20.5% increase in final leaf length. In addition, no significant difference was observed between transgenic and non-transgenic siblings for the maximal LER at steady-state growth in independent experiments (Supplementary Table 2; Fig. 4a,b). To address this puzzle, we determined the leaf elongation duration (LED)[25] which is defined as the time interval during which the leaf grows from 100 mm until fully grown. The LED was significantly and reproducibly increased in all three *GA2ox:PLA1* transgenic lines as compared with their non-transgenic siblings (Fig. 4a, Supplementary Table 2) and this was also reflected in their LER profiles (Fig. 4a, Supplementary Fig. 8). Since the size of the division zone size was not consistently significantly different (Supplementary Table 2, Fig. 4a), we tested if *GA2ox:PLA1* promotes cell division by extending the period in which the division zone remains active, by following the growth of leaf four and the size of the division zone over time (Fig. 4b). In the *GA2ox:PLA1-P3* transgenic plants, during steady-state growth, the size of the division zone was slightly but not always significantly larger compared with the non-transgenic siblings (Fig. 4b), in accordance with the kinematic analysis

(Supplementary Table 2). However, towards the end of steady-state growth, the LER and size of the division zone remained significantly higher in the *GA2ox:PLA1* plants compared with the non-transgenic siblings (Fig. 4b). The interaction between genotype and LER over time was highly significant (two-way mixed model, $P = 0.0175$, $n = 3$), as well as the interaction between genotype and division zone size over time (two-way ANOVA, $P = 0.021$, $n = 3$). The more pronounced difference in leaf growth between the transgenic and non-transgenic siblings during the later phases of leaf growth was supported by the observation that leaf length became significant ($P = 0.003$, mixed model analysis with custom hypothesis Wald tests (corrected for multiple testing), $n = 3$) at the time point where LER started to decrease (day six in Fig. 4b,c).

A homozygous *pla1 Mutator* transposon mutation (mu1044329), in which the transposon was at the border of the intron acceptor site and exon 2 was obtained (Supplementary Fig. 9) and the comparison to the wild type showed that the *pla1* mutant had contrasting phenotypes than when GA2ox:PLA1 was compared to its respective wild-type plants. In segregating and homozygous progeny, the observed phenotype always correlated to the presence of the transposon in *PLA1*. In the homozygous mutants the *PLA1* transcript levels could be detected when assayed with RT–qPCR using *PLA1*-specific primers upstream and downstream the Mu insertion (Supplementary Fig. 9); however, we were unable to amplify the region spanning the insertion in mutant cDNA. The *pla1* homozygous mutant plants displayed a 9.2% shorter final leaf length ($P = 0.02$, Student's $t$-test, $n > 20$),

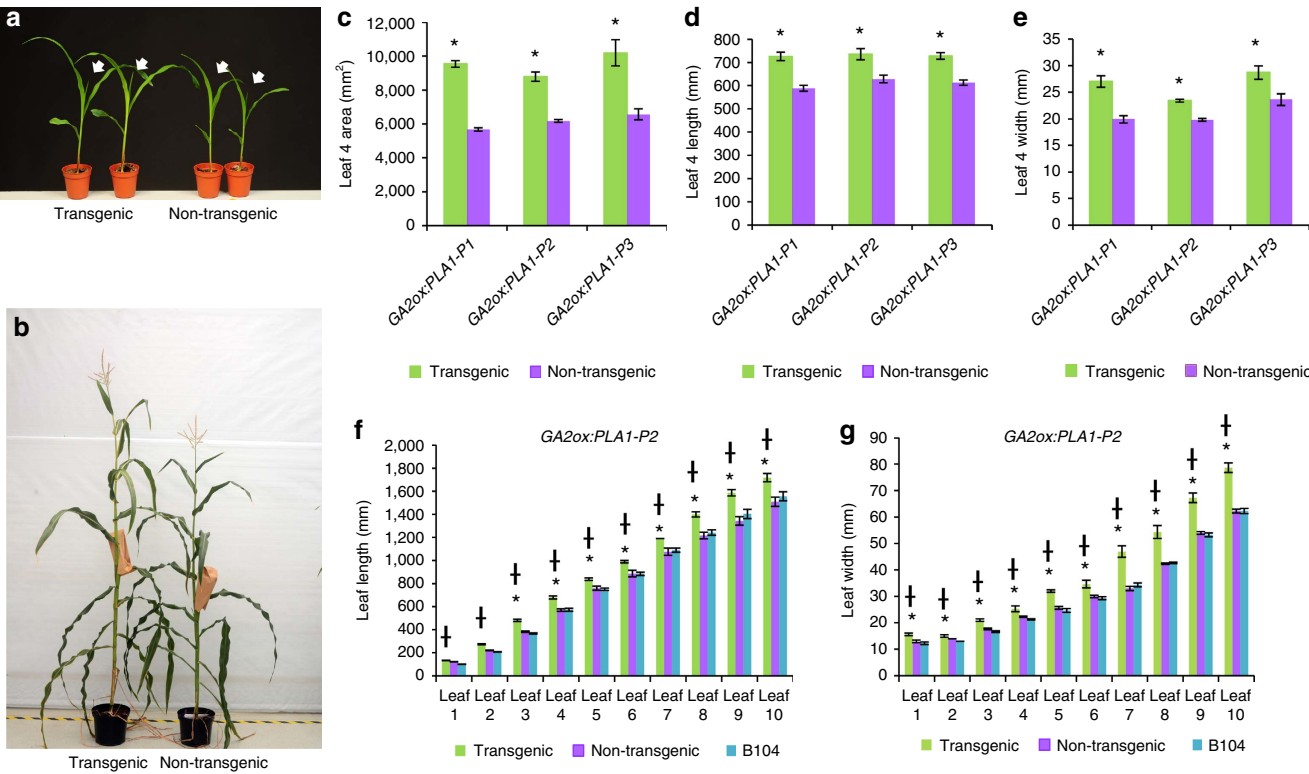

**Figure 2 | Phenotype of *GA2ox:PLA1* plants. (a)** Seedlings (12 days after sowing; white arrows indicate the newly appeared leaf four) and **(b)** mature plants (96 days after sowing) of the *GA2ox:PLA1-P3* segregating population. **(c–e)** Mature leaf four phenotypes in three independent *GA2ox:PLA1* segregating lines. Error bars indicate standard errors. Significant differences ($P < 0.05$, Student's *t*-test, $n = 5$) are indicated with an asterisk (*). **(f)** Leaf length of *GA2ox:PLA1-P2* compared with its non-transgenic siblings and B104 ($n = 3$). **(g)** Leaf width of *GA2ox:PLA1-P2* compared with its non-transgenic siblings and B104 ($n = 3$). **(f,g)** * indicates significant difference between *GA2ox:PLA1-P2* and its non-transgenic siblings; † between *GA2ox:PLA1-P2* and B104 ($P < 0.05$, mixed model analysis with custom hypothesis Wald tests (corrected for multiple testing), $n = 3$).

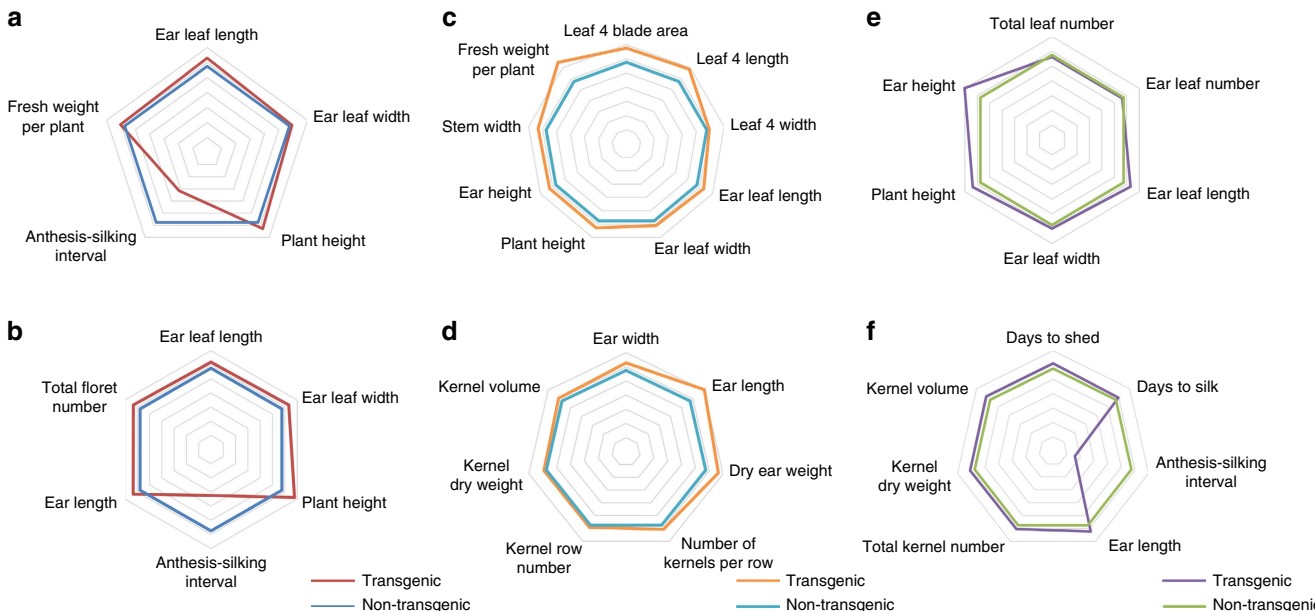

**Figure 3 | Phenotypic evaluations of the *GA2ox:PLA1-P2* transgenic line in the greenhouse and the field.** Phenotypes of the *GA2ox:PLA1-P2* plants in B104 background, relative to the segregating non-transgenic siblings in the greenhouse **(a)** and the US field trial 2014 **(b)**. Vegetative and reproductive phenotypes of *GA2ox:PLA1-P2* (T) X CML91 and the non-transgenic *GA2ox:PLA1-P2* (NT) X CML91 hybrids grown in the field at Belgium **(c,d)** and US **(e,f)** in 2015. The spider webs present phenotypic measurements of plants containing the *GA2ox:PLA1* construct relative to the non-transgenic controls (set at 100%).

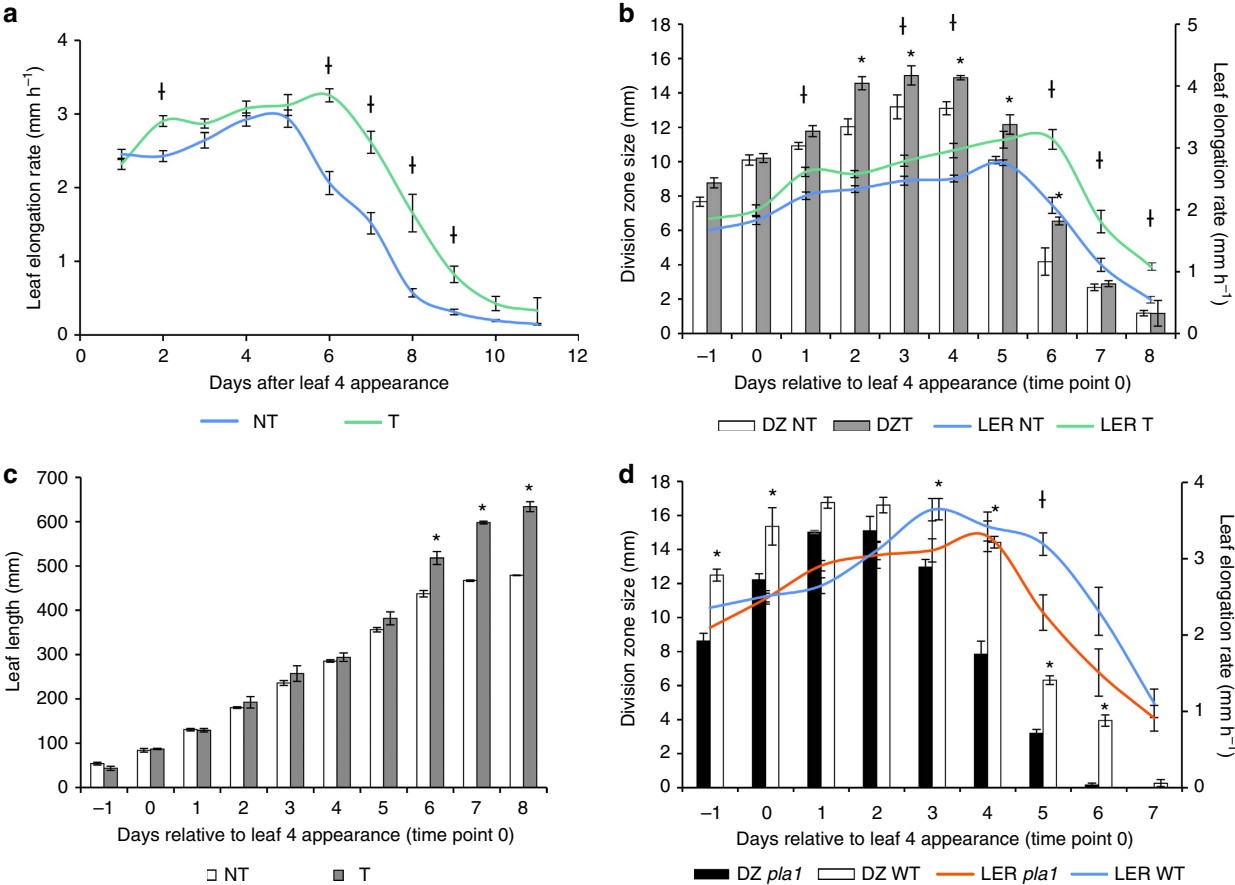

**Figure 4 | Leaf four growth phenotype of *GA2ox:PLA1-P3* and the *pla1* mutant over time.** (**a**) Leaf elongation rate (LER) of leaf four from day 1 until day 11 ($n = 6$). † indicates significant interaction between genotype and days ($P < 0.0001$, mixed model analysis with custom hypothesis Wald tests (corrected for multiple testing, $n \geq 5$). (**b**) Division zone size and LER measurements over time in segregating *GA2ox:PLA1-P3* from day $-1$ till day 8 ($n = 3$). (**c**) Leaf length measurements over time in segregating *GA2ox:PLA1-P3* ($n = 3$). * indicates a significant interaction between genotype and days ($P < 0.0001$, mixed model analysis with custom hypothesis Wald tests (corrected for multiple testing), $n = 3$). (**d**) Division zone size and LER measurements over time in the *pla1* mutant and wild-type (WT, $n = 3$). (**b,d**) * indicates a significant interaction between genotype and days for division zone size ($P = 0.021$, analysis of variance with custom hypothesis *t*-tests (corrected for multiple testing), $n = 3$), while † highlights a significant interaction between genotype and days for LER ($P = 0.0175$, mixed model analysis with custom hypothesis Wald tests (corrected for multiple testing, $n = 3$). Error bars indicate standard errors. NT, non-transgenics; T, transgenics.

an overall decrease in the size of the division zone and a premature decline of the maximal growth rate as compared with wild-type plants (Fig. 4d).

**Prolonged cell division duration as a compensatory mechanism.**
Phenotypic analysis showed that *PLA1* plays a role in setting the developmental time window for cell division and thus in maintaining steady-state growth over time. To determine how the duration of steady-state growth is related to maximal steady-state growth, the *GA2ox:PLA1* plants were challenged by conditions in which maximal steady-state growth was positively or negatively affected.

Previously, high levels of bioactive gibberellic acid (GA), resulting from ectopic expression of *UBIL:GA20ox-1*, were shown to increase the size of the division zone[13], while here we show that *PLA1* mainly expands the time window of cell division. Crosses were made between the segregating *UBIL:GA20ox-1* line and the segregating *GA2ox:PLA1-P2* line. Plants expressing both transgenes had both the high maximal LER of the *UBIL:GA20ox-1* transgene and the prolonged growth duration of the *GA2ox:PLA1* transgene (Fig. 5a), resulting in an increase in final leaf size and other phenotypic traits that were additive in

growth chamber as well as field grown plants (Fig. 5b,c, Supplementary Table 3).

Alternatively, it was shown that mild drought conditions (Supplementary Table 4) or cold nights[26] resulted in a comparable reduction in the LER. In the non-transgenic siblings, mild drought resulted in a significant reduction of LER (ranging from $-25.0$ to $-27.9\%$ reduction) which is partly compensated by a $13.6$–$18.2\%$ prolonged LED (Supplementary Table 4a). In the *GA2ox:PLA1* transgenic plants, the reduction in LER was slightly more severe (ranging from $-30.4$ to $-34.6\%$ reduction), while the compensation from the LED was more pronounced (ranging from $23.4$ to $28.8\%$) compared with the non-transgenic siblings (Fig. 5d, Supplementary Table 4a). In cold nights, a similar decrease in LER ($-25.6\%$ for *GA2ox:PLA1-P3*, $-25.4\%$ for non-transgenic siblings) was observed in both transgenic and non-transgenic plants, but in contrast to mild drought stress, the cold-induced reduced LER was not compensated by an extended LED in the non-transgenic siblings. However, a $9.8\%$ extended LED was observed in cold-treated *GA2ox:PLA1-P3* plants (Fig. 5e, Supplementary Table 4b). These data indicate that localized overexpression of *PLA1A* can partially compensate for growth reduction induced by adverse environmental conditions.

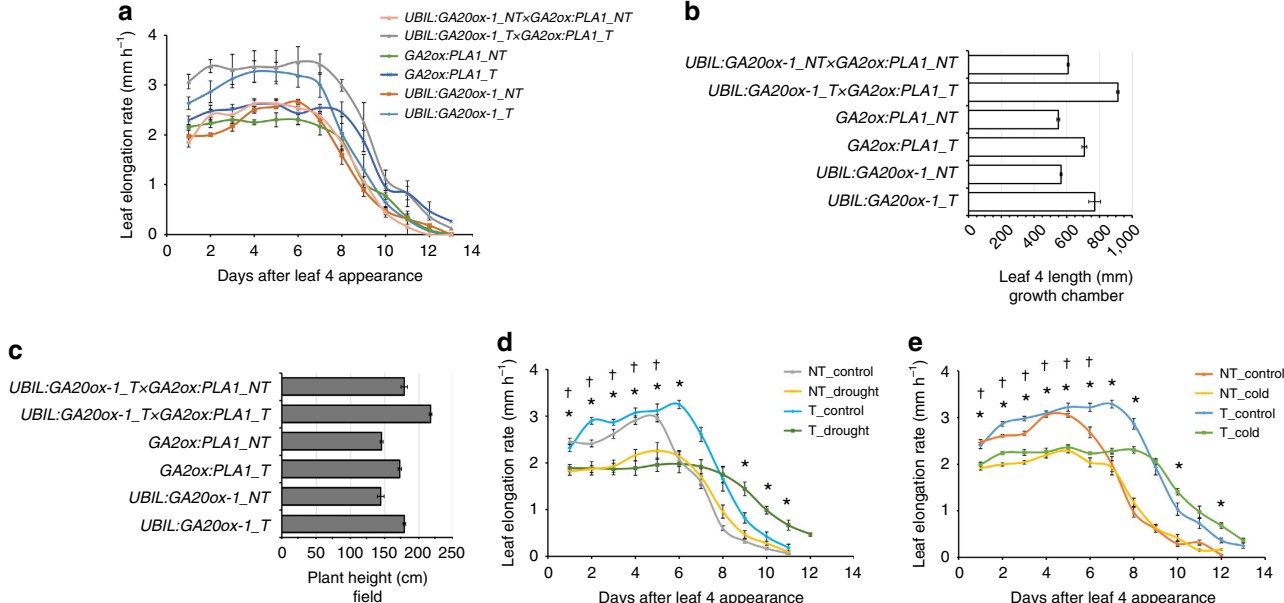

**Figure 5 | Phenotypes of *GA2ox:PLA1* in combination with *UBIL:GA20ox-1* and under cold nights or mild drought conditions.** (**a–c**) Phenotypes of *UBIL:GA20ox-1 × GA2ox:PLA1*: leaf elongation rate (**a**) and final length of leaf four (**b**) of *UBIL:GA20ox-1 × GA2ox:PLA1* grown in the growth chamber; final plant height (**c**) of *UBIL:GA20ox-1 × GA2ox: PLA1* grown in the US field. (**d**) Leaf elongation rate of *GA2ox:PLA1* under mild drought stress and (**e**) cold nights. * and † indicate significant differences between control and stress for the transgenics and non-transgenics, respectively ($P < 0.05$, mixed model analysis with custom hypothesis Wald tests (corrected for multiple testing), $n > 5$). Error bars indicate standard errors. NT, non-transgenics; T, transgenics.

**Ectopic expression of *PLA1* alters auxin metabolism.** To investigate the mechanisms underlying how *PLA1* affects the LED, the transcriptome of the most basal half centimetre of leaf four was profiled at two times: during steady-state growth (2 days after the appearance of leaf four) and when leaf length became significantly different between the *GA2ox:PLA1-P3* transgenic and non-transgenic siblings (6 days after the appearance of leaf four; Fig. 4b).

The transcriptomic changes (fold change > 2 and false discovery rate < 0.05) between time points (2,444 and 1,845 transcripts in non-transgenic and *GA2ox:PLA1* transgenics, respectively) were much more pronounced than between the genotypes (115 and 553 transcripts at day 2 and day 6, respectively). The highest upregulated transcript was *PLA1*, which was 51.8 and 47.5 times higher than the endogenous *PLA1* transcript level at day 2 and day 6, respectively.

The genes that were downregulated in both genotypes over time were enriched for the GO categories 'cell cycle arrest', 'auxin metabolic process' and 'maintenance of meristem identity' (Supplementary Table 5a–b). Conversely, the genes upregulated over time were enriched for the GO categories 'photosynthesis', 'response to light' and 'auxin mediated signalling pathway' (with nine *SMALL AUXIN-UP RNA* (*SAUR*) genes and genes involved in auxin efflux) (Supplementary Tables 5a–b, 6). These data suggest that some differentiation is already taking place in the division zone as leaf growth progresses.

When the difference in the size of the division zone was maximal between transgenic and non-transgenic siblings (at day 6, Fig. 4c), the genes with a higher expression level in the transgenic plants were enriched for the GO categories 'negative regulation of DNA binding transcription factor activity' and 'regulation of hormone levels', while the downregulated genes were enriched for 'terpenoid catabolic process' and 'auxin polar transport' (several auxin efflux transporters including *PIN1b*, *PIN10a* and *BIF2*, a positive regulator of cellular auxin efflux; Supplementary Tables 5d, 6).

To further analyse the effect of ectopic *PLA1* expression on auxin metabolism, the levels of auxin, precursors and conjugates were determined at 0.5 cm sampling intervals along 3 cm of the growth zone during steady-state growth. In the growth zone of *GA2ox:PLA1-P3* plants, auxin levels as well as the levels of auxin biosynthesis precursors such as tryptophan (TRP) and indole-3-pyruvic acid (IPyA) as well as auxin conjugates (IAA-Glu; IAA-Asp) and the auxin inactivation product (oxIAA) were consistently higher than those of the non-transgenic siblings (Fig. 6). This increase in TRP, IAA and auxin conjugates was also observed in an independent experiment performed on centimetre 1 and centimetre 3 of the growth zone (Supplementary Fig. 10). The increased accumulation of auxin was confirmed by the elevated levels of the auxin response reporter DR5rev:mRFPer (ref. 27) that were observed in the basal half centimetre of the growing fourth leaf by fluorescence quantification and RT–qPCR (Supplementary Fig. 11).

**Discussion**

Leaf growth is driven by cell division and cell expansion and both processes are regulated in space and time[28]. In the maize leaf, the mechanisms that were shown to affect growth are mainly involved in the spatial organization of the growth zone. The size of the division zone and the maximal growth rate is determined by the levels of GA as shown by the changes in division zone size, LER and final leaf length in the GA biosynthetic and signalling mutants and *UBIL:GA20ox-1* overexpression lines[13]. In addition to GA, mutations in *BRASSINOSTEROID INSENSITIVE1* (ref. 15) and *GROWTH-REGULATING FACTOR1* (ref. 14), and stress growth conditions have been shown to affect maize leaf growth by altering the division zone size (Supplementary Table 4)[29]. Here, we identified a role for *PLA1* in the temporal regulation of maize leaf growth. Both constitutive and localized overexpression of *PLA1* resulted in longer leaves with longer growth periods (and LED) compared with their

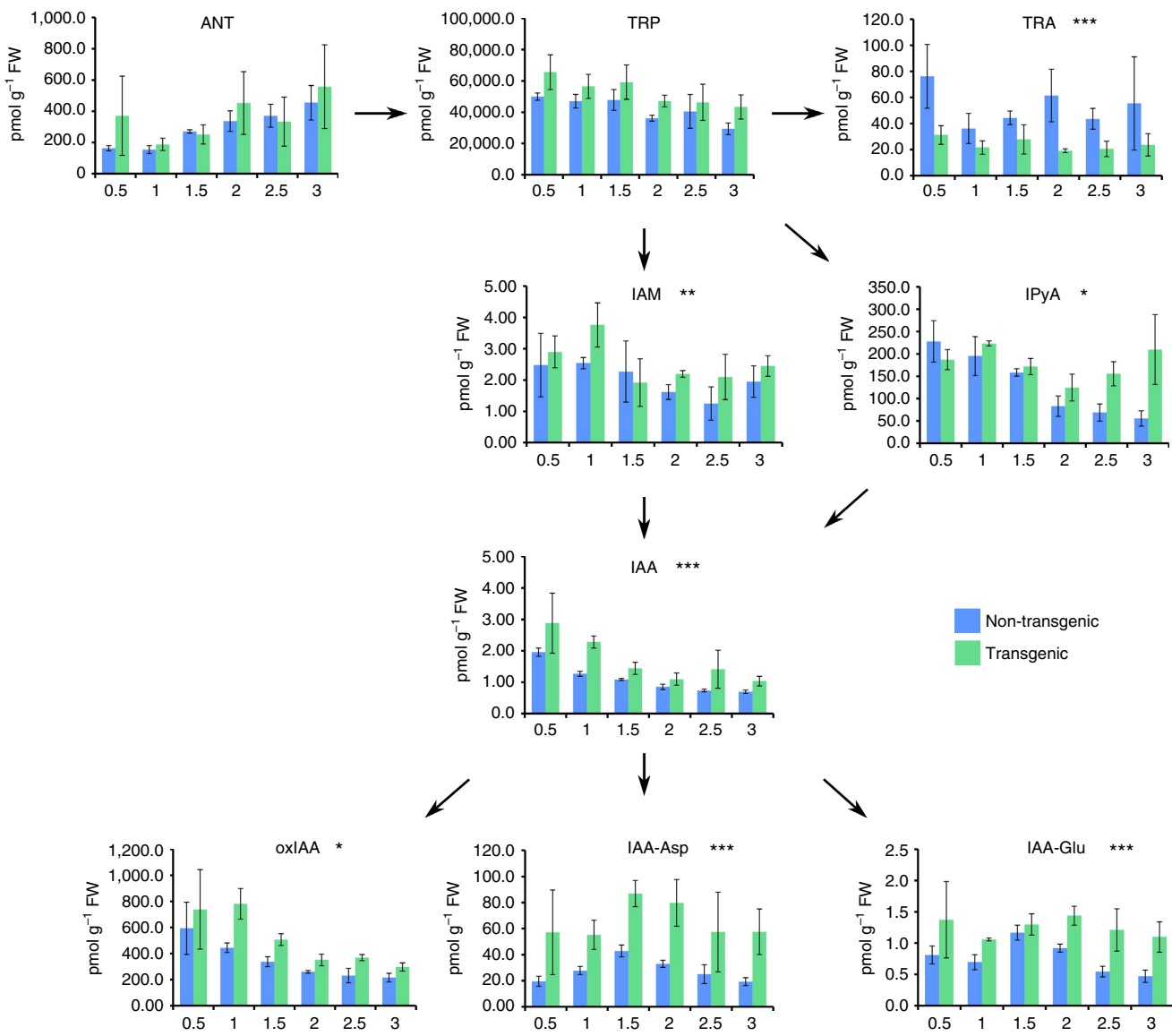

**Figure 6 | Bioactive IAA level including IAA precursors and inactivation products in *GA2ox:PLA1-P3*.** *, ** and *** indicate significant differences between genotypes with *P* values < 0.05, < 0.01 and < 0.001, respectively, mixed model analysis with custom hypothesis Wald tests (corrected for multiple testing) (n = 3). Error bars indicate standard errors. ANT, anthranilate; IAA-Asp, IAA-aspartate; IAA-Glu, IAA-glutamate; IAA, indole-3-acetic acid; IAM, indole-3-acetamide; IPyA, indole-3-pyruvic aicd; oxIAA, 2-oxoindole-3-acetic acid; TRA, tryptamine; TRP, tryptophan.

controls. The phenotypes of the stacked transgenics containing *UBIL:GA20ox-1* and *GA2ox:PLA1* were additive compared with the single transgenes, suggesting that in certain situations, LER and LED can be decoupled and that PLA1 is mainly regulating LED. A similar conclusion was reached when natural variation for LER and LED was studied in the B73xH99 recombinant inbred line (RIL) population, as both processes were highly correlated with final leaf length but not with each other[30,31].

Interestingly, at least two hormones play a role in the spatial and temporal regulation of maize leaf growth. GA promotes leaf growth by increasing the maximal growth rate (LER)[13], whereas auxin is downregulated over time in wild type and increased auxin levels in the *GA2ox:PLA1* transgenic lines are associated with a prolonged LED. In accordance with previous study[13], we show here that higher levels of auxin accumulate in the basal region of the division zone, which decrease to lower levels at the distal boundary of the division zone. This pattern is parallel to the expression profile of *PLA1*. Interestingly, overexpression of

the auxin-induced *AUXIN REGULATED GENE INVOLVED IN ORGAN GROWTH* (*ARGOS*) gene results in enlarged leaves and a delayed flowering time[32], similar to *GA2ox:PLA1* lines. Additionally, chorismate synthase that catalyses a reaction resulting in the auxin precursor chorismate[33] as well as the auxin transporter *ZmPIN1b* (ref. 34) are downregulated in *GA2ox:PLA1* compared with the non-transgenic siblings after the steady-state growth stage, and are negatively correlated with LED (ref. 30). Together, these data suggest that auxin may play a central role in determining the duration of maize leaf growth. Also PLA1, a molecular player in the LED process was linked to auxin, as the moderate overexpression of PLA1, that resulted in growth and yield enhancement, had higher auxin levels at the leaf base and increased levels of the auxin response marker DR5rev:mRFPer. The actual substrate and product, catalysed by *PLA1* is still unknown, but there are no indications so far that *PLA1* catalyses a step in the biosynthesis of auxin. *PLA1* could, however, influence auxin concentration by producing a molecule that

inhibits IAA oxidase or by modulating auxin transport. The identification of the nature of the PLA1 enzymatic products might provide insights how PLA1 is able to affect auxin metabolism.

We generated two PLA1 overexpression lines (UBIL:PLA1 and GA2ox:PLA1), both of which increased the length and width of the leaves, which was more pronounced in UBIL:PLA1 than in GA2ox:PLA1. Constitutive overexpression of PLA1 results in very small mature cells which might be due to over-proliferation of cells that fail to expand. In addition to members of the CYP78A family, several other growth-regulating proteins also overstimulate cell division and delay differentiation[35]. When CYCD3;1 is overexpressed in Arabidopsis, the mature leaf epidermis of the transgenic plants consists of a large number of very small cells due to overproliferation[36]. Additionally, overexpression of the cell cycle regulator E2Fα and their dimerization partner DPα severely affects plant stature due to cellular overproliferation[37]. In these cases, growth stimulation is highly dependent on the expression level of growth regulators. Therefore, using tissue-specific or inducible promoters to drive gene expression instead of constitutive promoters can overcome such unfavourable effects. The use of specific promoters to drive gene expression has recently been shown to positively influence agronomically important traits[4,38], and the specific expression of PLA1 provides another example of the importance of promoter choice in the design of transgenic constructs.

Targeted overexpression of PLA1 not only positively affected leaf and plant size and stover yield, but also seed yield. The increase in kernel number and kernel weight resulted in a significant increase in final grain production. In Arabidopsis, elevated PLA1 levels results in bigger seeds by promoting cell proliferation of the integument[18]. Overexpression of the tomato PLA1 homologue increases fruit size by producing more cells in pericarp and septum tissues[39]. Here, we demonstrated that tissue-specific but spatially expanded expression of the PLA1 gene results in a simultaneous increase in both stover and seed yield.

Recently, several growth-related genes were successfully introduced into different genetic backgrounds in maize. Maize ARGOS and GROWTH-REGULATING FACTOR10 maintained their growth-regulating capacity in one tested hybrid background[3,40], while ectopic overexpression of the rice trehalose-6-phosphate phosphatase in maize displayed drought tolerance in several hybrid backgrounds[4]. Similarly, GA2ox:PLA1 showed consistent, positive effects on plant growth and yield in different genetic inbred and hybrid backgrounds grown in both the greenhouse and two independent field trials that differed in growing season and geographical location. Such stability of the transgene's effect strengthens the potential utility of PLA1 to improve biomass and seed yield in other grain crops in addition to maize.

## Methods

**Phylogenetic analysis.** Predicted amino sequences for CYP78A-like genes were identified using BLAST from the genomes of representative land plant taxa (Arabidopsis thaliana, Aquilegia caerulea, Zea mays, Oryza sativa, Musa acuminata, Amborella trichopoda, Picea abies, Selaginella moellendorffii, Physcomitrella patens and Marchantia polymorpha). Sequences were aligned using the MUSCLE algorithm[41]. ProtTest3.4.2 (ref. 42) was used to determine the optimal model of amino acid substitution: LG + I + G + F. Bayesian phylogenetic estimation was performed with MrBayes3.2.6 (refs 43,44), using 1 million generations and a burnin of 25%.

**Maize transformation and genotyping.** The maize PLA1 or CYP78A1 (GRMZM2G167986) was amplified by extention overlap PCR. The PLA1 gene was driven by the UBIL promoter[45] or GA2-oxidase (GA2ox) promoter, that comprises of 2,046 bp upstream from the start codon of GA2-oxidase (GRMZM2G031724; GA2ox) and was ligated into the vector pBbm42GW7 (ref. 46) (http://gateway.psb.ugent.be/). Immature embryos of the maize inbred line B104

were transformed by Agrobacterium tumefaciens cocultivation[47]. For GA2ox:PLA1, we obtained ten independent events from transformation, of which the progeny of one event had a low germination rate. From the nine other lines, six were single locus lines for which only in three independent events the expression of the transgene could be shown (GA2ox::PLA1-P1, GA2ox::PLA1-P2 and GA2ox::PLA1-P3). Remarkably, the lines with an observable 'over'-expression of PLA1 were also the lines that displayed the growth phenotypes. We used these three independent events for phenotyping in the growth chamber and the greenhouse, but Belgian legislation did not permit cultivating lines that contain backbone vector DNA. Therefore, we were only able to perform the field trials on two independent events. Primary transgenic events in which the T-DNA was present in a single locus were backcrossed to B104 resulting in 1:1 segregation. These segregating plants were used for the growth chamber and greenhouse evaluations in order to exclude maternal effects. For field trials, both transgenic and non-transgenic plants of this segregating generation were selfed for two generations. The homozygous plants were crossed to CML91 to obtain transgenic and non-transgenic hybrids. The presence of the PAT protein was tested by an immunochromatographic assay (AgroStrip, Romer), leaf painting or PCR. Expression levels were monitored using RT-qPCR using 18S rRNA as housekeeping gene and the levels were determined using the $2^{-\Delta\Delta CT}$ method. The primers used in this study are summarized in Supplementary Table 7.

**Growth conditions in growth chamber and greenhouse.** Plants for leaf growth monitoring were grown under growth chamber conditions with controlled relative humidity (55%), temperature (24 °C day/18 °C night), and light intensity (170–200 µmol per m[2] per second photosynthetic active radiation at plant level) provided by a combination of high-pressure sodium vapour (RNP-T/LR/400W/S/230/E40; Radium) and metal halide lamps with quartz burners (HRI-BT/400W/D230/E40; Radium) in a 16 h/8 h (day/night) cycle. Plants under mild drought treatment shared the same growth conditions but the water content of the control condition was 1.23 g water g$^{-1}$ dry soil, and the water content of mild drought soil was 0.738 g water g$^{-1}$ dry soil, corresponding to −0.023 MPa and −1,025 MPa, respectively. For the cold treatment, plants were grown in the same growth conditions during the photoperiod but at 4 °C in the dark. A gradual decrease and increase of radiation intensity was implemented over 0.5 h to mimic dusk and dawn. Plants for adult plant trait characterization were grown under controlled greenhouse conditions (26 °C/22 °C, 55% relative humidity, light intensity of 180 µmol per m[2] per second photosynthetic active radiation, in a 16 h/8 h day/night cycle).

**Plant material and phenotype evaluation in growth chamber.** Plants were measured daily to determine the LER (n = 5). The leaf blade was cut and scanned to determine its leaf area by ImageJ (ref. 48). Kinematic analysis was performed based on Nelissen et al.[12]. To determine the division zone size over time, leaf four was harvested daily before emergence from the sheath of leaf three until fully grown. The time point was determined by the day when leaf four was initially visualized from the whorl of leaf three. The size of the division zone was determined by the distance between the base and the most distally observed mitotic cell in DAPI-stained leaves along the proximal-distal axis with a fluorescence microscope (AxioImager, Zeiss). For every analysis, at least three technical replicates were taken.

**Field trial design and plant trait analysis.** Field experiments were conducted in 2014 and 2015 at Ames, Iowa, US and in 2015 at Wetteren, Belgium. In US 2014, three GA2ox:PLA1 segregating families were grown in three rows, 20 plants per row (nine inches between plants), the plant traits were measured in all plants. In 2015, a GA2ox:PLA1-P2 segregating population and the F1 hybrid with CML91 containing the transgene (GA2ox:PLA1-P2 × CML91) were grown in five and ten rows, respectively: each row contained 20 plants (nine inch between plants), plant traits (plant height, ear leaf length, ear leaf width) and grain traits (anthesis silking interval, total floret number, ear length) were performed on 12 representative plants per row that were chosen by disregarding border and off-looking plants. In Belgium, two transgenic hybrids GA2ox:PLA1-P1 × CML91 and GA2ox:PLA1-P2 × CML91 and their non-transgenic controls were grown in randomized block design with three replicates with planting density over 88,000 plants per hectare. Each replicate contained four rows and 40 plants per row. All the plants were measured for vegetative trait analysis, except ear-leaf width and stem width were measured on 20 and five representative plants per replicate, respectively. Cob component data were determined by harvesting five representative ears per replicate. Because the legislation in Belgium requires detasseling of transgenic plants in the field, both the transgenic and non-transgenic plants were detasseled and pollination originated from B104xCML91 border plants. The necessity to detassel transgenic plants precluded any observation of anthesis, so that the anthesis silking interval could not be determined.

**RNA sequencing analysis.** Library preparation was done using the TruSeq RNA Sample Preparation Kit v2 (Illumina). In brief, poly(A)-containing mRNA molecules were reverse transcribed, double-stranded cDNA was generated and adapters were ligated. After quality control using a 2100 Bioanalyzer (Agilent), clusters were

generated through amplification using the TruSeq PE Cluster Kit v3-cBot-HS kit (Illumina), followed by sequencing on an Illumina NextSeq500 with the TruSeq SBS Kit v3-HS (Illumina). Sequencing was performed in paired-end mode with a read length of 75 nt. The quality of the raw data was verified with FastQC (http://www.bioinformatics.babraham.ac.uk/projects/fastqc/, version 0.9.1). Next, quality filtering was performed using the trimmomatic: reads were globally filtered so that for at least 75% of the reads, the quality exceeds Q10 and 3' trimming was performed to remove bases with a quality below Q10, ensuring a minimum length of 20 bp remaining. Reads were subsequently mapped to v3 of the maize B73 reference genome (http://ftp.maizesequence.org/B73_RefGen_v3/) using GSNAP 2.0.0 (ref. 49). The concordantly paired reads that uniquely map to the genome were used for quantification on the gene level with htseq-count from the HTSeq.py python package[50].

These genes were subjected to differential analysis with the R software package edgeR[51,52] (R version 3.2.3). Only genes with an expression value higher than 1 cpm (corresponding to five read counts) in at least three samples were retained for the analysis (20,284 genes were kept out of 39,323). Trimmed Mean of M-values normalization[53] was applied using the *calcNormFactors* function. Variability in the data set was assessed with an MDS plot. All three biological replicates clustered nicely together and there was a clear separation between the four factor level combinations. Trended negative binomial dispersion parameters were estimated with the default Cox–Reid method based on a model with main effects of treatment, time and replicate and an interaction term between time and treatment using the *estimateGLMTrendedDisp* function, followed by the estimation of the empirical bayes dispersion for each transcript using the *estimateGLMTagwiseDisp*. A negative binomial regression model was then used to model the over dispersed counts for each gene separately with fixed values for the dispersion parameter as outlined in ref. 54 and as implemented in the function *glmFit* using the above-described model. A likelihood ratio test (LRT) was performed to compare this model with a model without replicate to assess possible replicate (batch) effects. After false discovery rate adjustments of the P values with the method described in ref. 55, only eight genes were found to have a significant batch effect as expected by the MDS plot. The estimate of the dispersions and the fitting of the model was repeated with now only the main effects of time and treatment and their interaction. The significance of the interaction term was assessed with an LRT test comparing the full model with the main effects model. To test user-defined hypotheses, the model was re-parameterized. The factors were combined to one factor with four levels, and a no intercept single factor model was fitted to the data. With this design, dispersions were re-estimated and the model was refit. The four contrasts of interest were in the difference between the time points for each genotype, and the difference between the genotypes at each time point. Significance was assessed with an LRT test and as before, FDR adjustments of P values were applied. All edgeR functions were applied with default values.

**IAA metabolite profiling.** Leaf four of *GA2ox:PLA1-P3* transgenic plants and non-transgenic siblings were harvested at the second day after the appearance from the whorl to simultaneously profile the majority of known auxin precursors and conjugates by a mass spectrometry-based method[56,57]. Five plants were taken for one biological replicate, and three biological replicates were harvested. Samples (50 mg fresh weight) were homogenized and extracted in ice-cold 50 mM sodium phosphate buffer (pH 7) with the addition of the following [$^2$H]- and [$^{13}$C]-labelled internal standards: [$^{13}$C$_6$]-IAAsp, [$^{13}$C$_6$]-IAGlu, (5 pmol per sample); [$^{13}$C$_6$]-ANT, [$^{13}$C$_6$]-IAA, [$^{13}$C$_6$]-IAM, [$^2$H$_4$]-IPyA, [$^{13}$C$_6$]-oxIAA, [$^2$H$_2$]-TRA (10 pmol per sample); [$^2$H$_5$]-TRP (100 pmol per sample). The plant extracts were incubated at 4 °C with continuous shaking (15 min), centrifuged (15 min, 23,000g at 4 °C), and then divided in two halves. In one half, the pH was adjusted to 2.7 with 1 M hydrochloric acid, and the sample was purified by solid phase extraction using an Oasis HLB columns (1 cc per 30 mg, Waters, Milford, USA) conditioned with 1 ml methanol, 1 ml water and 0.5 ml sodium phosphate buffer (pH 2.7). After sample application, the column was washed with 2 ml 5% methanol and then eluted with 2 ml 80% methanol. The eluate was evaporated to dryness *in vacuo* and stored at − 20 °C until liquid chromatography/multiple reaction monitoring/ mass spectrometry (LC-MRM-MS) analysis. The second half of the supernatant (ca 0.5 ml) was derivatized by 3 ml of 0.25 M solution of cysteamine (adjusted with NH$_3$ to pH 8.0). The samples were incubated for 1 h at room temperature and purified as described above.

All samples were analysed by LC-MRM-MS. The evaporated samples were dissolved in 40 μl of 10% methanol before mass analysis using a 1290 Infinity Binary LC System coupled to the 6490 Triple Quad LC/MS System with Jet Stream and Dual Ion Funnel technologies in positive mode (Agilent Technologies). The samples were injected onto a reversed-phase column (Kinetex C18, 50 x 2.1 mm, 1.7 μm; Phenomenex) and separated using an 11-min gradient composed of 0.1% acetic acid in methanol (A) and 0.1% acetic acid in water (B) at a flow rate of 0.25 ml min$^{-1}$, column temperature of 30 °C, and a binary linear gradient: 0 min, 10:90 (A:B); 10.0 min, 50:50 (A:B); 11.0 min, 98:2 (A:B). At the end of the gradient the column was washed with 100% methanol (1 min) and re-equilibrated to initial conditions (3 min). Determination of endogenous auxins was performed by MRM of the protonated precursor and appropriate product ions. The MRM transitions, instrument settings, retention times and detection limits were optimized for each analyte[57]. The linear range spanned at least five orders of magnitude with a

correlation coefficient of 0.9985–0.9999. The MassHunter software (Version B.05.02, Agilent Technologies) was used to determine the concentration, using stable isotope dilution.

**Quantification of DR5rev:mRFPer expression.** Confocal images of the basal half cm of leaf four of *GA2ox:PLA1-P3 × DR5rev:mRFPer* (ref. 58) maize plants, segregating for the *PLA1* transgene, were acquired using an inverted Zeiss710 CLSM microscope. Leaves were mounted between slide and coverslip in water. Objective used was the Plan-Apochromat 20 × /0.8 Dry. mRFP was excited with the 561 nm laser line and the emission was detected between 580 and 640 nm. Per plant five places in the leaf were randomly chosen and all images (z-stacks) were acquired using identical settings. For quantification, all images were analysed using the ImageJ software (Rasband, W.S., ImageJ, US NIH, Bethesda, Maryland, USA, http://imagej.nih.gov/ij/, 1997–2012). Confocal images were converted to 8-bit depth images. Analysis was done on single slices with a thickness of 0.78 μm. Since the *DR5rev:mRFPer* expression in the leaf was maximal in the cells surrounding the xylem, the xylem cells were taken as a reference point to select single slices for analysis. The average mRFP intensity of the top 1% highest pixels of the phloem to xylem ratio was calculated in Microsoft Excel and statistically compared between transgenic and non-transgenic plants for the *PLA1* transgene by a two sample *t*-test.

**Statistical methods.** Leaf area measurements of leaf nine (UBIL:PLA1, Fig. 1b), leaf four (GA2ox:PLA1, Fig. 2c–e) and epidermal cell length (UBIL:PLA1, Fig. 1d) were analysed with two-tailed Student's *t*-tests to compare the mean in the transgenic line with a control line using the t.test function from the R software[52]. Student's *t*-tests were also performed on data presented in Supplementary Tables 1,2,4; Supplementary Figs 3,7,10.

Measurements over time on a transgenic line and non-transgenic line (such as division zone size measurements and leaf length) were submitted to an analysis of variance (Fig. 4b–d). At the different time points, different plants were measured. These data were modelled with a general linear model in SAS with the fixed main effects time and genotype and their interaction term. The interest was in the difference between the transgenic line and the non-transgenic line at each day. These comparisons were estimated with Wald-type tests using the plm procedure. *P* values were adjusted for multiple testing using Sidak's adjustment method as implemented in the multtest procedure from SAS. This analysis will be referred to as ANOVA.

The measurements of leaf length and leaf width on a series of leaves originating from the same plant are longitudinal in nature (Fig. 2f,g). A mixed model was fitted to leaf length and leaf width with the mixed procedure from SAS (Version 9.4 of the SAS System for windows 7 64 bit Copyright 2002–2012 SAS Institute Inc. Cary, NC, USA, www.sas.com). The correlation between measurements done on the same plant is accounted for by modelling the variance-covariance matrix of the residuals. Several structures were tested: unstructured, (heterogenous) compound symmetry, (heterogenous) autoregressive and (heterogenous) banded toeplitz. The best structure was chosen based on AIC values. For the mean model, a model with the fixed main effects genotype and leaf and their interaction term was compared with a linear spline model with a knot at leaf two. The latter model was chosen based on AIC values. This model contained the fixed effects genotype, leaf, leaf2 (representing the truncated basis) and the interaction effect between leaf and genotype. To speed up calculations, Fisher scoring was used in the first step of maximum likelihood estimation. The Kenward-Roger approximation for computing the denominator degrees of freedom for the tests of fixed effects was applied. The interest was in the difference between the transgenic line compared to B104 and the non-transgenic line at each leaf. These comparisons were estimated with Wald-type tests using the plm procedure. *P* values were adjusted for multiple testing using the MaxT method as implemented in SAS. This method controls the family wise error rate precisely at the significance level α (here set at 0.05) and is not conservative. In the absence of a significant interaction, as was the case for the leaf width measurements (Fig. 2f), the difference between the transgenic line compared to B104 and the non-transgenic line is the same at each leaf. In this case, *P* values were corrected with the Dunnett adjustment method as implemented in SAS. Width measurements were log2 transformed prior to analysis. This analysis will be referred to as leaf series analysis.

LER in the different lines was measured over time on the same plants. These experiments are thus also longitudinal in nature. The analysis was performed in the same way as for the leaf series, except that both genotype and day were considered as categorical variables. The interest was in the difference between the transgenic line and the non-transgenic line at each day. These comparisons were estimated with Wald-type tests using the plm procedure. *P* values were adjusted for multiple testing using the MaxT method as for the leaf series (Fig. 4a,b,d and Supplementary Fig. 8a,b). In the absence of a significant interaction term, only the difference between the transgenic line and the non-transgenic (averaged over the days) was estimated. In the drought experiments, the fixed effects part of the model for LER (Fig. 5d,e) contained three main effects: genotype, treatment and days together with all possible higher order terms. Here, the interest was in the difference between the drought and control treatment for each genotype and at each day. This analysis will be referred to as LER analysis.

Levels of auxin related metabolites were measured in different zones of the leaves of three transgenic plants and three non-transgenic plants (Fig. 6). Since the measurements done in the different zones of a leaf, these data are also longitudinal in nature and was analysed the same way as the LER. A significant interaction between position in the zone and genotype was only detected for IPyA. At each zone, the difference in IPyA levels between the genotypes was estimated with a Wald test. P values were adjusted for multiple testing using the MaxT method as before. For IAM, IAA, OXIAA, IAA-Asp and IAA-Gly, only the main effects were significant. In the absence of an interaction effect, the difference between the genotypes is the same in all zones. This difference was estimated with a Wald test. For TRA, only a significant main genotype effect was detected. The difference between the transgenic line and non-transgenic line was again estimated with a Wald-type test. For ANT and TRP, only a significant zone effect was detected.

For all analyses, residual diagnostics were carefully examined. A comparison was declared significant when the adjusted P value was smaller than 0.05.

**Data availability.** RNA-seq data are available in the ArrayExpress database (www.ebi.ac.uk/arrayexpress) under accession number E-MTAB-5422. The authors declare that all other data supporting the findings of this study are available within the manuscript and its supplementary files or are available from the corresponding author on request.

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

## Acknowledgements

The research leading to these results has received funding from the European Research Council under the European Community's Seventh Framework Programme (FP7/2007-2013) under ERC grant agreement no. (339341-AMAIZE)11, from Ghent University ('Bijzonder Onderzoeksfonds Methusalem project' no. BOF08/01M00408 and Multidisciplinary Research Partnership 'Biotechnology for a Sustainable Economy' Grant 01MR0510W), and from the Interuniversity Attraction Poles Programme (IUAP P7/29 'MARS') initiated by the Belgian Science Policy Office. K.L. would like to acknowledge the support from the Swedish Foundation for Strategic Research (VINNOVA) and the Swedish Research Council (VR) and K.L. and O.N. thank the Swedish Metabolomics Centre for the use of instrumentation. X.S. is funded by a PhD-grant from the Chinese Scholarship Council (CSC). C.W. was supported by funding from the National Science Foundation (IOS-1253421). The authors highly appreciated the help of all field trial volunteers and of René Custers for the field trial application. We thank Frederik Coppens for the logistics concerning RNA sequencing, Daniel Van Damme for help with the confocal image analysis, and Annick Bleys for the critical reading of the manuscript and the precious help to improve and finalize it.

## Author contributions

X.S., H.C., D.I. and H.N. designed the study. X.S., K.F., T.V.H., C.W., S.D., C.V., H.N., K.D., J.D.B. performed the experiments. J.C. carried out and analysed the field data in the US under the supervision of M.M.; A.D.V. and H.N. directed and X.S., K.F., T.V.H., C.V., K.D., J.D.B., A.D.V., H.N. together with some volunteers performed field observations, monitoring and measurements for the field trail in Belgium. X.S. and H.N. analysed the field data. X.S. and T.V.H. performed the RNA sequencing analysis. V.S. provided statistical support. O.N. and K.L. performed metabolite profiling. G.C. and M.V.L. did the maize transformations. T.V.H. performed the confocal imaging and quantification of the DR5 marker. X.S., D.I. and H.N. wrote the manuscript.

## Additional information

**Competing financial interests:** The authors declare no competing financial interests.

