## [Peer Review File · Nature Communications]

Reviewers' comments:

Reviewer #1 (Remarks to the Author):

The manuscript entitled "Altered expression of maize KLUH enhances biomass and seed yield by extending cell division" describes the development genetically modified maize that extend the expression of the KLUH gene using the GA2ox promoter. This significantly increases leaf growth and some yield components. The results show that this is due to a slightly higher and longer duration of cell division in the leaf. This is an outstanding study demonstrating that GM technology can have a positive impact on important crop productivity components.

Of note in this paper is a nice demonstration showing the impact of trait gene expression control on physiological readout. First, they demonstrate that use of a strong constitutive promoter, Ubi1, which is common practice in many academic labs produces infertile plants that make large leaves. They solve this problem by limiting transgene expression to a slightly wider range of the leaf cell division zone using the GA2ox promoter. This is an important result that shows the impact of transgene expression control. They also report plant characterization work that took place in both controlled and field environments. In this case, they show that observations in controlled environments extend to the field. However, the field studies do not reflect production environments because planting density is too low. In addition the authors evaluated the trait effect in both the transformed inbred, B104, and up to 5 hybrid backgrounds.

The weaknesses in this paper are mainly in the detail in several figure legends, a lack of detail in the Methods and accounting for the statistical analysis in some of the comparisons. Also, the number of transgenic events produced for this study is on the low end. Typically for GM yield traits, more than 5 and ideally 10 independent events are required to clearly demonstrate GM trait efficacy. Finally, the language in the text and several figures need to be reviewed and edited for flow and agreement.

The Figure 2 legend suggests that white arrows should also be in 2b. in addition, the description includes results of a 'linear spline mixed model' in two places that has nothing to do with the data in the figure since the indicators of statistical significance are the result of a separate test described at the end of the legend. What's confusing is why the different tests were done, which should be clarified. This also applies to Figure 4. In Figure 3, the language is difficult to follow since it describes results from three different experiments, each with different materials. The sample sizes are not indicated and it's also not clear if any of the contrasts are significant. Figures 5 and 6 would also benefit from a more detailed accounting of the statistical tests. My suggestion is to add a Statistics section to the Methods.

The Methods section lacks detail in several places. The PCR steps used to produce the KLUH gene could be more detailed since not all readers of this journal are going to be skilled molecular biologists. In line 330, the use of 'cloned' to describe a ligation step can be confusing to some. The primer orientation should be indicated. In line 333-334, the scope of

the GA2ox promoter is not described properly. The last sentence of the first section, lines 340-42, lists primers to determine the KLUH expression level. How was this actually measured? In lines 347-350, PCR is not a method used to assay protein levels. In the last part of this section, the details of the thermal cycling protocol are given, but there are no details regarding the PCR reactants. In the 'RNA sequencing analysis' section, it's not clear what 'bayes' means.

In the 'Growth conditions in growth chamber and greenhouse' section, the growth conditions between night and day should be consistently detailed. Also, the units for light intensity should be micromole per m² per sec. The imposed drought conditions are also not described well. What do the kPa terms refer to? In the next section, line 387, visible seems to be a better term than visualized.

The 'Field trial design and plant trait analysis section' needs the most improvement. How were the seeds used in these studies produced? Since the paper includes studies using several different materials it's important to document their relationship to each other. What are the dimensions of the rows? Are there alleys in between rows? Particularly with corn, plant density is known to influence productivity, something the paper tries to address. What is meant by the term 'segregating'? How were the 12 representative plants in each row selected? The trait characterization measurements should be more detailed, including a description of the logic for the different replication strategies.

Specific Comments on the text:

In Results, line 74 the transformed maize are referred to as 'plants' which is too generic. Lines or events are a better term.

In Results, line 78 the 'control transformant' is a little confusing. Is it an escape that lacks a T-DNA insert or has a partial T-DNA insert?

The paragraph beginning with 'The enhanced leaf phenotypes persisted in hybrids...' is very difficult to follow. Rather than describe aspects of several different experiments, the authors should describe each one in succession. In addition hybrids are often referred to as the F1 generation.

In the second paragraph of the 'KLUH stimulated growth by extending the duration of cell division' section, the referenced figure should be Supplementary Fig. 9.

In the second paragraph of the 'KLUH-mediated growth stimulation involves auxin metabolism' section, FC and FDR should be spelled out. Also, I am not sure what is meant by 'the division zone becomes consumed' in the last sentence in paragraph 2.

In the Discussion section, line 256, the reference is probably 24 not 25.

In the Discussion section, line 259, the term stacked may be better than doubled.

In the Discussion section, line 270, the reference is not clear.

In the Discussion section, line 276, the word catalyzes seems misplaced.

In the Discussion section, lines 284-286, a better way to write this sentence is 'Constitutive overexpression of KLUH results in very small mature cells which might be due to over-proliferation, producing cells that fail to expand.'

In the Discussion section, lines 290-293, it's not clear what DP-alpha is or why it's relevant.

In Supplementary Figure 1a what are the expression data relative to?

In Supplementary Figure 3, it's difficult to believe that many of the significant differences indicated are real given the size of the standard error bars. It's likely that the control data were also variable.

In Supplementary Figure 4 could be more detailed since this study included several hybrids which many investigators would consider to be different entries.

In Supplementary Figure 6 it would be good to indicate the statistical test used.

In Supplementary Figure 8 it would be nice to understand the rationale for the statistical tests, as in Figures 2 & 4.

Supplementary Table 1 shares data the Supplementary Figure 4 and only 4 plants were used to produce the data. In Sup. Fig. 4 sample sizes range from 16-23. Why the difference? More detail will help the table stand alone, such as which leaf is being compared.

In Supplementary Table 3 it would be useful to add a column indicating source of each comparison.

In Supplementary Table 4 the statistical tests are potentially confusing. The comparison appears to be between the two treatments and not between transgenic and non-transgenic plants.

In Supplementary Table 5 the use of W6 and W2 and K2 and K6 to designate the entries makes the information a little difficult to follow.

Reviewer #2 (Remarks to the Author):

In this study, Sun et al., characterize the effects of ZmKLUH/CYP78A1 expression on plant growth and yield. The authors observed that overexpressing KLUH with a GA2ox promoter, known to be expressed specifically in expanding cells, produced plants with an overall

increase in size and yield, overcoming infertility derived from ubiquitous overexpression of this gene. The effects of this specific overexpression were stable in various genetic backgrounds and growing locations/seasons. The authors show that KLUH is involved in the temporal regulation of maize leaf growth by influencing the duration of the cell division period, also known as leaf elongation duration (LED). The authors suggest that this expansion of the LED might be connected with auxin regulation and independent of GA. Additionally, they suggest that KLUH overexpression might be used to overcome the growth inhibition caused by abiotic stresses like cold or drought.

Although these observations are interesting and deserve attention, in my opinion, the study does not provide enough novel insight on the mechanisms behind KLUH function to merit publication in Nature Communications. The observation that KLUH regulates organ size in a way that positively correlates with its expression level is not novel and has been described in multiple species, as mentioned by the authors. It is also known that Cytochrome p450 proteins can affect cell size by regulating the duration of the cell division period. The more general aspects of the data that show a connection between auxin and duration of LED, and that LER and LED are independent are also not novel, as referenced in the manuscript. Furthermore, the study fails to provide insight on the connection between KLUH and auxin metabolism, and for the compensatory effects of KLUH overexpression under stress conditions.

Specific comments:

The description of the effects of KLUH overexpression on plant growth is extensive and the results appear consistent through multiple experiments. Nevertheless, the way the data is presented often appear redundant or was confusing. For example, Fig3, SuppFig3, SuppFig4 and SuppTable1 all have related data but is presented in various formats and is partially redundant. I had a really hard time understanding the figures and getting a good overview of the differences between experiments, and of the different transgenic lines.

There are also several small inconsistencies in the data presented, that make it harder to interpret. For example, regarding Fig1 and SuppFig1, in the text, it is said the increase in leaf size correlates with the expression level of KLUH. Nevertheless, P1 is expressed at higher levels than P2, but makes smaller leaves. In the same figures, error bars and statistical tests are missing (also absent in other figures like SuppFig2b). Are the differences between P1 and P2 significant? Furthermore, there is an inconsistency between the leaf analyzed in each experiment, which adds to the confusion: expression of *kluh* was analyzed in leaf 8, leaf area determined in leaf 9 and cell length measured in leaf 4, for which there is no leaf size data.

I think it would be interesting to see how the stress conditions actually affect the phenotypes that were measured extensively before, regarding plant size, biomass, yield... That way it would be possible to address if KLUH overexpression can actually be useful to overcome negative effects of stress on plant growth.

The authors don't provide enough information about the *kluh* mutant. Mutant plants should

be molecularly characterized and a reference or more information about their origin should be provided. I also think it would be interesting to do a more detailed analysis of the mutant plants. SuppFigure9 is another example of the lack of statistical analysis.

The role of auxin in KLUH function is not discussed conveniently. The data showing changes in auxin-related genes and analysis of auxin levels is presented but no real conclusion is made regarding KLUH function.

Although GA2ox::KLUH-P3 is expressed much higher than the other transgenes, the impact on plant size is not more pronounced. Nevertheless, both LER and size of division zone seemed higher on GA2ox::KLUH-P3 but the description in the text is very vague and the difference between the transgenic lines is not properly discussed. This also makes it harder to understand the choice of P3 to perform the detailed analysis of the division zone over time.

Overexpression of KLUH seems to have a negative impact on leaf number but that is not really explained.

In some experiments, the size of the samples analyzed was small. For example, in SuppTable1 n=4.

Line 147: there is a wrong reference to Fig4a,b. The steady state analysis referenced is in Supptable2 (only for P1 and P2).

Line 151: there is a reference to Supptable2 has having data for GA2ox::KLUH-P3, but that table only has values for the other two lines. P3 is only analyzed in Fig4a,b. Where are the values of LED for P3?

Line 170: wrong reference to SuppFig8. It should be SuppFigure9.

Reviewer #1 (Remarks to the Author):

The manuscript entitled “Altered expression of maize KLUH enhances biomass and seed yield by extending cell division” describes the development genetically modified maize that extend the expression of the KLUH gene using the GA2ox promoter. This significantly increases leaf growth and some yield components. The results show that this is due to a slightly higher and longer duration of cell division in the leaf. This is an outstanding study demonstrating that GM technology can have a positive impact on important crop productivity components.

Of note in this paper is a nice demonstration showing the impact of trait gene expression control on physiological readout. First, they demonstrate that use of a strong constitutive promoter, Ubi1, which is common practice in many academic labs produces infertile plants that make large leaves. They solve this problem by limiting transgene expression to a slightly wider range of the leaf cell division zone using the GA2ox promoter. This is an important result that shows the impact of transgene expression control. They also report plant characterization work that took place in both controlled and field environments. In this case, they show that observations in controlled environments extend to the field. However, the field studies do not reflect production environments because planting density is too low.

We realize that our field trials might not reflect actual production sites in the USA, but we took, as many agronomical characteristics as possible into account. For the field trial in Belgium, we sew with a density of approximately 89.000 seeds per hectare and given the very high germination rate, the planting density was higher than 85.000 plants per hectare. In Belgium, the planting density varies for commercial varieties depending on variety and sowing date, between 80.000 and 100.000 plants per hectare, indicating that our planting density is within the range of what is commonly used in agriculture. The planting density was added to the Methods section.

In addition the authors evaluated the trait effect in both the transformed inbred, B104, and up to 5 hybrid backgrounds. The weaknesses in this paper are mainly in the detail in several figure legends, a lack of detail in the Methods and accounting for the statistical analysis in some of the comparisons.

We removed the explanation of the statistical analyses from the figure legends and we added a topic covering all statistical analyses in the Methods sections. As a result, the figure legends now read more fluently and the reasoning behind the statistical analyses is better explained.

Also, the number of transgenic events produced for this study is on the low end. Typically for GM yield traits, more than 5 and ideally 10 independent events are required to clearly demonstrate GM trait efficacy.

We agree with the reviewer that more independent event are better for field evaluations. We obtained ten independent events from transformation, of which the progeny of one event had a low germination rate. From the nine other lines, six were single locus lines for which only in three independent events the expression of the transgene could be shown (GA2ox::PLA1-P1, GA2ox::PLA1-P2 and GA2ox::PLA1-P3). Remarkably, the lines with an observable ‘over’-expression of PLA1 were also the lines that displayed the growth phenotypes (Figure 1). We used these three independent events for phenotyping in the growth chamber and the greenhouse, but the Belgian legislation did not allow for cultivating lines that contained backbone vector DNA. Therefore, we were only able to

perform the field trials on two independent events. This information was added to the Methods section.

Figure 1. *KLUH* expression relative to 18s ribosomal RNA for at the basal 0.5 cm region of the fourth leaf of four *GA2ox::KLUH/PLA1* transgenic (T) lines and their non-transgenic (NT) siblings. Error bars indicate standard errors (n=3). Asterisks indicate statistically significant difference in the transgenic plants versus the non-transgenic plants according to a Student's t-test (* and ** correspond to P-value of $0.05 > P > 0.01$ and $0.01 > P > 0.001$, respectively).

Finally, the language in the text and several figures need to be reviewed and edited for flow and agreement.

We improved the language throughout the text and the figure legends were simplified by describing the statistics in a designated section of the Methods.

The Figure 2 legend suggests that white arrows should also be in 2b.

We adjusted the figure legend accordingly so the text explaining the white arrows is only mentioned for Figure 2a.

in addition, the description includes results of a 'linear spline mixed model' in two places that has nothing to do with the data in the figure since the indicators of statistical significance are the result of a separate test described at the end of the legend. What's confusing is why the different tests were done, which should be clarified. This also applies to Figure 4. In Figure 3, the language is difficult to follow since it describes results from three different experiments, each with different materials. The sample sizes are not indicated and it's also not clear if any of the contrasts are significant. Figures 5 and 6 would also benefit from a more detailed accounting of the statistical tests. My suggestion is to add a Statistics section to the Methods.

We followed the suggestion of the reviewer and provided a statistics section in the Methods. The different tests were performed because the nature of the collected data differed. A Student's t-test was used to compare quantitative end-point measurements (e.g. final leaf length, plant height, number of kernels, etc.) between transgenic and non-transgenic plants. Different methods were used when parameters were followed over time, depending on the sampling strategy. The size of the

division zone over time was determined by a destructive method (fixing and staining the basal part of the leaf for microscopical analysis), meaning that different plants were used on the different time points. To determine the difference between the transgenic and non-transgenic plants at each time point, the data were modeled using a general linear model. The measurements of leaf length over time, were non-destructive and were therefore performed on the same plants over time. Since these data were longitudinal in nature, a mixed model was used to fit the data. The data in the figures don't change, but the legends no longer contain the information on the statistical analyses.

The Methods section lacks detail in several places. The PCR steps used to produce the KLUH gene could be more detailed since not all readers of this journal are going to be skilled molecular biologists. In line 330, the use of 'cloned' to describe a ligation step can be confusing to some. The primer orientation should be indicated. In line 333-334, the scope of the GA2ox promoter is not described properly. The last sentence of the first section, lines 340-42, lists primers to determine the KLUH expression level. How was this actually measured? In lines 347-350, PCR is not a method used to assay protein levels. In the last part of this section, the details of the thermal cycling protocol are given, but there are no details regarding the PCR reactants. In the 'RNA sequencing analysis' section, it's not clear what 'bayes' means.

We adjusted the Methods section according to the reviewer's comments and we added a supplemental table listing all primers, with the orientation.

In the 'Growth conditions in growth chamber and greenhouse' section, the growth conditions between night and day should be consistently detailed. Also, the units for light intensity should be micromole per m² per sec. The imposed drought conditions are also not described well. What do the kPa terms refer to? In the next section, line 387, visible seems to be a better term that visualized.

This Methods sections was more detailed and the spelling errors were corrected.

The 'Field trial design and plant trait analysis section' needs the most improvement. How were the seed used in these studies produced? Since the paper includes studies using several different materials it's important to document their relationship to each other. What are the dimensions of the rows? Are there alleys in between rows? Particularly with corn, plant density is known to influence productivity, something the paper tries to address. What is meant by the term 'segregating'? How were the 12 representative plants in each row selected? The trait characterization measurements should be more detailed, including a description of the logic for the different replication strategies.

We adjusted the methods section to address these issues raised by the reviewer. Our transformation pipeline typically transforms B104 plants and the transformed events are backcrossed to B104, resulting in a progeny that consists of 50% transgenics and 50% non-transgenics, when the T-DNA is inserted in a single locus. We prefer to work with these seed stocks, that we refer to as segregating, for the phenotypic analysis in the growth chamber and the greenhouse, as the non-trangenics have gone through the same tissue culture steps as the transgenics and the seeds originated from the same cob, eliminating seed batch effects. However, to proceed to field trials we are not allowed to genotype (e.g. by leaf painting) the plants but we are obliged to know the genotype in advance. For this reason, we selfed transgenic and non-transgenic plants from a segregating population until we obtained homozygous plants in the T3 generation. We experienced that B104 was not a suitable genetic background for field trials in the temperate climate in Belgium (Voorend et al., 2016, Plant Biotech J.), so we evaluated the performance in Belgium field trials of different hybrids between B104 and early flowering inbreds (2013-2015). Because the hybrid between B104 and CML91 grew

well, resulted in proper seed yield and was not prone to lodging in three consecutive field trials in Belgium, we crossed the homozygous transgenic and non-transgenic plants to CML91, resulting in transgenic (hemizygous) and non-transgenic hybrids, that were evaluated in the field trials mentioned in the manuscript. Because event P3 showed the presence of some vector backbone, beyond the T-DNA to be transferred to the plant DNA, this line was not allowed for field trial. The crossing scheme described above was performed for events P1 and P2.

In Belgium, the field trial was designed to have three plots per genotype that were randomly organized (so twelve in total; Figure 2). Each plot consisted of four rows of six meter each. Because the plants needed to be emasculated, B104xCML91 hybrids were sown in between the plots to assure pollination. No paths were made between the rows, but some small paths were formed between each block of three plots, to allow accessibility of the plots.

In the USA, three rows (each row is 14.6 feet and 35 inches apart) per genotype were (one kernel every 9 inch) sown and the twelve plants were selected by disregarding border plants and off-looking plants.

Figure 2. Field trial design of the of the field trial in 2015 in Belgium. T = transgenic, NT = non-transgenic.

Specific Comments on the text: In Results, line 74 the transformed maize are referred to as ‘plants’ which is too generic. Lines or events are a better term.

We replaced the word ‘plants’ by events.

In Results, line 78 the ‘control transformant’ is a little confusing. Is it an escape that lacks a T-DNA insert or has a partial T-DNA insert?

The control transformant is a plant containing a promoter::GUS construct, with no observable phenotype, that was chosen because it had a similar age as the UBIL::KLUH events, and had gone through the tissue culture steps. Besides the B104 plants, we included the control transformant to show that plants that went through tissue culture and regeneration following the maize transformation, were not responsible for the observed phenotypes. We performed this control because we realize early on that these plants might not produce viable seeds. However, since there is no significant difference between the control transformant and the B104 plants, we decided to omit this control and only mention the differences relative to B104, that are sufficiently convincing on their own.

The paragraph beginning with ‘The enhanced leaf phenotypes persisted in hybrids...’ is very difficult to follow. Rather than describe aspects of several different experiment, the authors should describe each one in succession. In addition hybrids are often referred to as the F1 generation.

In an attempt to describe the results as condensed as possible, this paragraph indeed got difficult to follow. We rephrased this paragraph and described the experiment successively. However, the data of the two field data at the two different locations (Iowa and Belgium) were introduced in parallel because the data were very similar and risk to be redundant if mentioned consecutively.

In the second paragraph of the ‘KLUH stimulated growth by extending the duration of cell division’ section, the referenced figure should be Supplementary Fig. 9.

We adjusted the text accordingly.

In the second paragraph of the ‘KLUH-mediated growth stimulation involves auxin metabolism’ section, FC and FDR should be spelled out.

We explained that FC is fold change and FDR is false discovery rate.

Also, I am not sure what is meant by ‘the division zone becomes consumed’ in the last sentence in paragraph 2.

In Figure 4b, it is illustrated that the size of the division zone gradually decreases over time and thus that the tissue at the base of the leaf becomes differentiated over time and is unable to maintain its dividing potential. The RNAseq data show molecularly that processes associated with differentiation rather than cell division (such as photosynthesis and cell wall biosynthesis). We used the terminology ‘consumed’ to describe that the division zone gets depleted over time, but we now removed part of the sentence mentioning ‘consumed’ as it might be confusing or unclear.

In the Discussion section, line 256, the reference is probably 24 not 25.

The reviewer is correct, we adjusted the text accordingly.

In the Discussion section, line 259, the term stacked may be better than doubled.

We adjusted the text accordingly.

In the Discussion section, line 270, the reference is not clear.

This refers to a manuscript from our team that was recently submitted to Plant Biotechnology Journal. A copy of the manuscript will be made available to the reviewers. According to the policy of the journal, papers that are not yet accepted can not be in the reference list.

In the Discussion section, line 276, the word catalyzes seems misplaced.

We rephrased the sentence to 'chorismate synthase that catalyzes a reaction resulting in the auxin precursor chorismate'.

In the Discussion section, lines 284-286, a better way to write this sentence is 'Constitutive overexpression of KLUH results in very small mature cells which might be due to over-proliferation, producing cells that fail to expand.'

We implemented the suggestion of the reviewer.

In the Discussion section, lines 290-293, it's not clear what DP-alpha is or why it's relevant.

DP is a transcription factor that heterdimerizes with E2F- α to regulate genes involved in cell cycle progression. We adjusted the text to make this more clear.

In Supplementary Figure 1a what are the expression data relative to?

The data in Supplementary Figure 1a are represented relative to the position that displayed the highest expression of a given gene, which is set at '1'. The graph is adjusted to show the highest expression of both genes at '1'.

In Supplementary Figure 3, it's difficult to believe that many of the significant differences indicated are real given the size of the standard error bars. It's likely that the control data were also variable.

Supplementary Figure 3 is indeed very complex and we originally added the data for completeness, but since the data in this figure add no novel data that is not presented in other figures or tables, we opted to omit this figure.

In Supplementary Figure 4 could be more detailed since this study included several hybrids which many investigators would consider to be different entries.

Supplementary Figure S4 represents the field trial data of the GA2OX::PLA1_P2 transgenic line in B104 inbred background in the USA in 2014. At that time we did not yet generate hybrids, so all data presented are in the same genetic background.

In Supplementary Figure 6 it would be good to indicate the statistical test used.

The statistical test was a Student's t-test. This information was added in the Figure legend.

In Supplementary Figure 8 it would be nice to understand the rationale for the statistical tests, as in Figures 2 & 4.

In supplemental Figure 8 the Leaf length is followed over time in a non-destructive way, so a mixed model was used to fit the data (as described above). Depending on the nature of the data represented in Figure 2 and Figure 4, leaf phenotypes were monitored over time, but depending on the sampling strategy (destructive or not), the same plants or other plants were measured at different time points, which dictates the statistical test used (see comment above).

Supplementary Table 1 shares data the Supplementary Figure 4 and only 4 plants were used to produce the data. In Sup. Fig. 4 sample sizes range from 16-23. Why the difference? More detail will help the table stand alone, such as which leaf is being compared.

Supplementary Table 1 summarizes the data on the transgenes in different hybrid backgrounds. From the cross with H99, few plants germinated (both in the combination with the transgenics as the non-transgenics). Therefore, the minimal number of observations was four so the legend should have stated $n \geq 4$. For the other combinations more individuals were tested, so the text is now adjusted to $n = 4-19$. There is no link between Supplementary Table 1 and Supplementary Figure 4: In the table we list the leaf phenotype of the PLA1 transgenes in the different hybrid backgrounds, showing that the phenotypes persist in different genetic backgrounds, while the figure illustrates the US field trial with the GA2OX::PLA1 line in an inbred background. Since this is not clear to the reviewers, we elaborated this paragraph to make the distinction between the different experiments.

In Supplementary Table 3 it would be useful to add a column indicating source of each comparison.

Supplementary Table 3 only shows the raw data of the stacked transgenes and does not represent comparisons.

In Supplementary Table 4 the statistical tests are potentially confusing. The comparison appears to be between the two treatments and not between transgenic and non-transgenic plants.

We changed the format of the table so it now allows for visualization of the differences between the transgenic and non-transgenic plants.

In Supplementary Table 5 the use of W6 and W2 and K2 and K6 to designate the entries makes the information a little difficult to follow.

The nomenclature was removed from Supplementary Table 5 and 6 and replaced by the full description of the samples.

Reviewer #2 (Remarks to the Author):

In this study, Sun et al., characterize the effects of ZmKLUH/CYP78A1 expression on plant growth and yield. The authors observed that overexpressing KLUH with a GA2ox promoter, known to be expressed specifically in expanding cells, produced plants with an overall increase in size and yield, overcoming infertility derived from ubiquitous overexpression of this gene. The effects of this specific overexpression were stable in various genetic backgrounds and growing locations/seasons. The authors show that KLUH is involved in the temporal regulation of maize leaf growth by influencing the duration of the cell division period, also known as leaf elongation duration (LED). The authors suggest that this expansion of the LED might be connected with auxin regulation and independent of GA. Additionally, they suggest that KLUH overexpression might be used to overcome the growth inhibition caused by abiotic stresses like cold or drought.

Although these observations are interesting and deserve attention, in my opinion, the study does not provide enough novel insight on the mechanisms behind KLUH function to merit publication in **Nature Communications**. The observation that KLUH regulates organ size in a way that positively correlates with its expression level is not novel and has been described in multiple species, as mentioned by the authors. It is also known that Cytochrome p450 proteins can affect cell size by regulating the duration of the cell division period. The more general aspects of the data that show a connection between auxin and duration of LED, and that LER and LED are independent are also not novel, as referenced in the manuscript. Furthermore, the study fails to provide insight on the connection between KLUH and auxin metabolism, and for the compensatory effects of KLUH overexpression under stress conditions.

We are convinced that the manuscript offers many novel insights. It was not yet shown that altered expression of a family member results in both stover and seed yield. Moreover, the choice of the promoter is based on the prior art information from our research on the maize growth zone. Previously, our research group indeed showed that LER and LED were two independent processes driving maize leaf growth by analyzing more than ten leaf and seedling parameter in a panel of recombinant inbred lines (RILs) (Baute et al., 2015). However, this study did not allow to identify a molecular player for LED, while in the submitted manuscript, we demonstrate that KLUH/PLA1 is a key regulator of the process. The additive phenotype of the stacked transgenes GA20OXIDASE and KLUH/PLA1 now proves genetically that the lack of correlation between LER and LED, observed by Baute et al., 2015, is indeed biologically relevant and points towards two independent processes. The method how we can study LED, by following the size of the division zone over time, is for the first time illustrated in this paper and should in future kinematic analyses be considered as an additional aspect. The submitted manuscript also shows a biological role for LED since this represents a compensatory mechanism to maintain growth for a longer period of time when conditions are suboptimal as was exemplified by both cold nights and mild drought. Finally, we identified that the KLUH/PLA1 mechanism functions through the elevated auxin levels. We now added additional data showing that the increased auxin levels are confirmed by elevated DR5rev::mRFP_{er} auxin response in the transgenes (both at cell biological and transcriptome level).

In summary, the manuscript shows that knowledge driven design of the transgene results in increased maize yield both in Europe and the USA. In addition, the manuscript provides evidence that PLA1 is the first identified molecular player to affect LED in maize. The LED mechanism is not only

independent of the GA driven LER leaf growth mechanism, but involves another phytohormone, auxin and plays a role in a compensatory growth mechanism under adverse conditions, that is different from what was previously described in Arabidopsis.

Specific comments:

The description of the effects of KLUH overexpression on plant growth is extensive and the results appear consistent through multiple experiments. Nevertheless, the way the data is presented often appear redundant or was confusing. For example, Fig3, SuppFig3, SuppFig4 and SuppTable1 all have related data but is presented in various formats and is partially redundant. I had a really hard time understanding the figures and getting a good overview of the differences between experiments, and of the different transgenic lines.

We realize that in an attempt to show all data, we confused the reader. For simplicity, and because the other reviewer also had issues with Supplementary Figure 3, we removed this figure from the manuscript. Supplementary Table 1 shows the confirmation of the leaf phenotypes in the different hybrids grown in the growth chamber, while Figure 3 and Supplementary Figure 4 represent a number of phenotypes that were scored in both greenhouse and field trials. The data in Figure 3 all describe the phenotypes of the transgenes introduced in a hybrid background, while Supplementary Figure 4 shows the data for the transgene in the B104 inbred background. Because this was not evident to understand, we adjusted the text to clarify these issues.

There are also several small inconsistencies in the data presented, that make it harder to interpret. For example, regarding Fig1 and SuppFig1, in the text, it is said the increase in leaf size correlates with the expression level of KLUH. Nevertheless, P1 is expressed at higher levels than P2, but makes smaller leaves. In the same figures, error bars and statistical tests are missing (also absent in other figures like SuppFig2b). Are the differences between P1 and P2 significant? Furthermore, there is an inconsistency between the leaf analyzed in each experiment, which adds to the confusion: expression of *kluh* was analyzed in leaf 8, leaf area determined in leaf 9 and cell length measured in leaf 4, for which there is no leaf size data.

The fact that we only obtained two independent events for the constitutive overexpression lines which were not able to result in progeny, impede proper statistical analysis. Indeed, RT-qPCR showed that the levels of overexpression might seem to be higher in the line with the 'smaller' leaf, but since we can not take several independent samples to obtain standard deviation, we can not be completely certain that this represents the actual situation. This being said, the phenotype of P1 was definitely much more pronounced than that of P2, because the plant's life cycle was more affected. Supplementary Figure 2 b and c (previously Supplementary Figure 1 b and c) shows that P1 produced much less leaves than P2. Therefore, the comparison of leaf nine is maybe not correct as the leaves might represent different developmental stages, but due to the fact that the two events were infertile we unfortunately could not perform a comparison based on developmental stage rather than time after sowing. Because the data for the constitutive overexpression lines only rely on two events, we based our conclusion on the dose-dependent effect of the transgene on the three independent events that we obtained for the moderate overexpression lines, on which proper

statistics could be performed. In addition, the differences between the constitutive and moderate overexpression of PLA1 also provide evidence for the dose-dependent nature of the gene function. However, showing the dose-dependency is not a primary aim of this paper, and therefore we removed the dose-dependent statements in the manuscript.

The lack of statistical power for the constitutive overexpression events also explains the other observations of the reviewer, namely that we can not conclude if the differences between P1 and P2 are significant. In addition, we realized that the chances to obtain progeny from these lines was limited already when the plants were growing, so we decided to perform as many analyses as possible on the two plants, resulting in the fact that the analyses were all performed on different leaf numbers. We took samples for RT-qPCR from the eighth leaf and we decided to take the unscathed leaf nine for the picture for publication. On the other hand, we performed a detailed analysis of leaf 4 for the moderate overexpression lines for which we got enough progeny.

I think it would be interesting to see how the stress conditions actually affect the phenotypes that were measured extensively before, regarding plant size, biomass, yield... That way it would be possible to address if KLUH overexpression can actually be useful to overcome negative effects of stress on plant growth.

We fully agree that it would be scientifically interesting to follow all mature phenotypes under drought stress and this will definitely be a scope of our future research. However, we used leaf 4 to examine the mechanisms of LER and LED and to unravel if the PLA1 gene is involved in the LED mechanism. So far, we do not have further insights in the exact reason why plants opt to grow for a longer period of time under adverse conditions nor what is the correlation, if any, between leaf LED and more mature phenotypes. This is an active field of research in our team and we hope to resolve these issues in the near future.

The authors don't provide enough information about the kluh mutant. Mutant plants should be molecularly characterized and a reference or more information about their origin should be provided. I also think it would be interesting to do a more detailed analysis of the mutant plants. SuppFigure9 is another example of the lack of statistical analysis.

The mutant originates from the Uniform-MU collection and harbors the transposon at the transition between the acceptor site of the intron and the second exon. We showed with Q-PCR that we were still able to detect transcript of the 5' and 3' part of the gene (so before and after the transposon insertion), albeit at a lower level as compared to the wild type plants. However, we are unable to amplify over the transposon insertion site on mutant cDNA. We added this information to the manuscript.

The role of auxin in KLUH function is not discussed conveniently. The data showing changes in auxin-related genes and analysis of auxin levels is presented but no real conclusion is made regarding KLUH function.

We added a paragraph in the discussion, speculating on how KLU/PLA1 can affect auxin levels.

Although GA2ox::KLUH-P3 is expressed much higher than the other transgenes, the impact on plant size is not more pronounced. Nevertheless, both LER and size of division zone seemed higher on GA2ox::KLUH-P3 but the description in the text is very vague and the difference between the

transgenic lines is not properly discussed. This also makes it harder to understand the choice of P3 to perform the detailed analysis of the division zone over time.

Originally, when we obtained the transgenic events, P3 looked like the best candidate for subsequent in detail analyses, because it had the highest expression level. The effect on LED and the size of the division zone is clearly and significantly present in P3 (Figure 4). We rephrased this paragraph to make the description of the phenotype less vague.

Overexpression of KLUH seems to have a negative impact on leaf number but that is not really explained.

That is a correct observation, however, our data (from greenhouse and different field trials) were not consistently pointing in that direction, which is why we opted not to focus on leaf number as a phenotype that would be affected by the transgene.

In some experiments, the size of the samples analyzed was small. For example, in SuppTable1 n=4.

Supplementary Table 1 summarizes the data on the transgenes in different hybrid backgrounds. From the cross with H99, few plants germinated (both in the combination with the transgenics as the non-transgenics). Therefore, the minimal number of observations was 4 so the legend should have stated $n \geq 4$. For the other combinations more individuals were tested, so the text is now adjusted to $n = 4-19$.

Line 147: there is a wrong reference to Fig4a,b. The steady state analysis referenced is in Supptable2 (only for P1 and P2).

The differences in steady state LER is represented in different ways throughout the manuscript: the LER calculated in Supplementary Table 2 of the kinematic analysis (P1 and P2) represents the average LER during the steady state growth, as described Nelissen et al., 2013. At the same time, the steady state growth is visualized when the LER is plotted over time as is visualized in Figure 4 (P3). We now refer to both the table as the graph as these together contain the information on the three independent events and thus show that the effect on LER and LED was present in the three independent events.

Line 151: there is a reference to Supptable2 has having data for GA2ox::KLUH-P3, but that table only has values for the other two lines. P3 is only analyzed in Fig4a,b. Where are the values of LED for P3?

The LED phenotype is significantly different for the 'weaker' overexpression events P1 and P2, as can be seen from Supplementary Table 2. We opted to visually represent the data for P3 as this is the event with the highest expression and the most pronounced phenotype. However, the numeric data of P3 for LED were also significant, which is indicated by the † sign in Figure 4. We just opted to not complicate Supplemental Table 2 even more by adding redundant information.

Line 170: wrong reference to SuppFig8. It should be SuppFigure9.

We adjusted the text accordingly.

REVIEWERS' COMMENTS:

Reviewer #1 (Remarks to the Author):

The manuscript entitled "Altered expression of maize PLASTOCHRON1 (ZmPLA1) enhances biomass and seed yield by extending the duration of cell division" describes the characterization of the maize ZmPLA1 gene (formerly (KLUH) primarily through the analysis of transgenic plants. The ZmPLA1 gene is over expressed using either a constitutive (Ubi1) or developmentally regulated (GA2ox) promoter. They also examine a pla1 mutant. Plant characteristics were evaluated in both controlled growth and field environments. A major discovery reported here is that extending ZmPLA1 expression using the GA2ox promoter significantly increases leaf growth and some yield components. The results show that this is primarily due to a longer duration of cell division in the leaf. The paper provides a detailed mechanistic analysis of the GA2ox::ZmPLA1 gene in maize. This is an outstanding study demonstrating that GM technology can have a positive impact on important crop productivity components. This report is distinct because (1) It focuses on how the transgene effect manifests at the cellular level. Many papers like this tend to focus on the productivity results which are often weak. (2) It connects a fundamental developmental process to maize productivity. (3) It demonstrates that ZmPLA1 exerts significant control on leaf cell division, however the biological mechanism was not fully worked out.

The present draft is much improved and resolves most of my concerns. The overall flow is excellent. The Methods, particularly their statistical analysis is much easier to follow (although I am not a statistician). My only major concern is their interpretation of the transcript profiling data. They use this evidence to connect ZmPLA1 with some aspects of auxin and cytokinin metabolism. I am not completely satisfied with their Gene Ontology analysis in Supplementary Table 5. In particular I do not see evidence for downregulation of 'cytokinin biosynthetic process' (line 228) in both genotypes. It only appears in 5a. Also I don't see evidence for upregulation of 'secondary cell wall biogenesis' (line 230) in either genotype. The evidence suggests that other GO terms like 'regulation of cellular biosynthetic process' for downregulation and 'carboxylic acid transport' for upregulation are affected in both genotypes. However these aren't mentioned in the text. This could be further clarified.

Minor comments:

Line 78-79: (first 0.5 cm) need only be mentioned once

Lines 83 and 90, indicate that B104 is wildtype when first mentioned

Line 120, Figure 3 represents one growing season, Supplementary Figure 4 represents 2 growing seasons

Line 123, replace B104 background with CML91 with B104xCML91 hybrid background

Line 153, remove 'size'

Line 169, It would be nice if this paragraph started with an explanation for the comparison to the pla1 mutant.

Lines 173-74, change 'when assays with RT-qPCR' to 'by RT-qPCR assays'

Lines 200 & 203, remove 'reduction'

Line 224, I could not find data mentioned

Line 325, It should be Fig, 4b

Line 249, Supplementary Figure 9 appears to be a subset of Figure 6. It's not clear why this is necessary.

Line 307, change statue to stature

Line 339, insert comma after mays

Line 349, change 'in 5' UTR' to 'upstream'

Line 350, add 'and was ligated into' after 'GA2ox)'

Lines 377 & 388, I am still not convinced that the stated light intensities are correct. For example full sunlight on a cloudless, clear day at high noon in the midwestern US is about 2000 $\mu\text{mol}/\text{m}^2/\text{s}$ PAR

Line 418, change 'replicates' to 'replicate'

Supplementary Table 6, column 4 the name is missing a P

Supplementary Figure 10, what is the difference between the two images in (c)?

Reviewer #2 (Remarks to the Author):

Thank you for addressing my comments regarding the manuscript. I believe the revised version is greatly improved in quality. I think the greatest improvements are in the simplification of the presented data and expansion of the material and methods section.

I do agree with the authors that the work presented is interesting and provides novel insights. Most importantly, it provides one more valid example of how biotechnology can be used to improve crop performance. This work is definitely relevant and deserves an audience.

I appreciate the limitations inherent to field trials and propagation of materials described by

the authors, but I still think the manuscript lacks the consistency in the data and novelty required for publication in Nature communications.

REVIEWERS' COMMENTS:

Reviewer #1 (Remarks to the Author):

The manuscript entitled “Altered expression of maize PLASTOCHRON1 (ZmPLA1) enhances biomass and seed yield by extending the duration of cell division” describes the characterization of the maize ZmPLA1 gene (formerly (KLUH) primarily through the analysis of transgenic plants. The ZmPLA1 gene is over expressed using either a constitutive (Ubi1) or developmentally regulated (GA2ox) promoter. They also examine a pla1 mutant. Plant characteristics were evaluated in both controlled growth and field environments. A major discovery reported here is that extending ZmPLA1 expression using the GA2ox promoter significantly increases leaf growth and some yield components. The results show that this is primarily due to a longer duration of cell division in the leaf. The paper provides a detailed mechanistic analysis of the GA2ox::ZmPLA1 gene in maize. This is an outstanding study demonstrating that GM technology can have a positive impact on important crop productivity components.

This report is distinct because (1) It focuses on how the transgene effect manifests at the cellular level. Many papers like this tend to focus on the productivity results which are often weak. (2) It connects a fundamental developmental process to maize productivity. (3) It demonstrates that ZmPLA1 exerts significant control on leaf cell division, however the biological mechanism was not fully worked out.

The present draft is much improved and resolves most of my concerns. The overall flow is excellent. The Methods, particularly their statistical analysis is much easier to follow (although I am not a statistician). My only major concern is their interpretation of the transcript profiling data. They use this evidence to connect ZmPLA1 with some aspects of auxin and cytokinin metabolism. I am not completely satisfied with their Gene Ontology analysis in Supplementary Table 5. In particular I do not see evidence for downregulation of ‘cytokinin biosynthetic process’ (line 228) in both genotypes. It only appears in 5a. Also I don’t see evidence for upregulation of ‘secondary cell wall biogenesis’ (line 230) in either genotype. The evidence suggests that other GO terms like ‘regulation of cellular biosynthetic process’ for downregulation and ‘carboxylic acid transport’ for upregulation are affected in both genotypes.

However these aren’t mentioned in the text. This could be further clarified.

Originally, the data were processed using version 2 of the B73 genome and were incorporated in the PhD thesis of Xiaohuan Sun, first author of the manuscript. For the publication of the data, we re-analyzed the data using version 3 of the B73 genome. This resulted in a slight difference in the significance of the GO-categories and, as the referee correctly pointed out, we indeed overlooked that the text and the table were not entirely aligned. The discrepancy between the text and the table is resolved in the new version of the manuscript.

Minor comments:

Line 78-79: (first 0.5 cm) need only be mentioned once

The text was adjusted accordingly.

Lines 83 and 90, indicate that B104 is wildtype when first mentioned

The changes were implemented.

Line 120, Figure 3 represents one growing season, Supplementary Figure 4 represents 2 growing seasons

Supplementary Fig 4 was added to the statement of the different growing seasons.

Line 123, replace B104 background with CML91 with B104xCML91 hybrid background

The suggested changes were made.

Line 153, remove 'size'

The word 'size' was removed.

Line 169, It would be nice if this paragraph started with an explanation for the comparison to the pla1 mutant.

The text was adjusted to explain the comparison to the pla1 mutant.

Lines 173-74, change 'when assays with RT-qPCR' to 'by RT-qPCR assays'

The text was adjusted accordingly.

Lines 200 & 203, remove 'reduction'

The word 'reduction' was removed twice.

Line 224, I could not find data mentioned

The data in paragraph starting with 'The transcriptomic changes' were only presented textually and not in any of the figures.

Line 325, It should be Fig, 4b

The reference to Fig.4b was made.

Line 249, Supplementary Figure 9 appears to be a subset of Figure 6. It's not clear why this is necessary.

The two figures represent data for two completely independent experiments and serve as a confirmation of the data. Originally we performed the metabolite measurements on six half centimeter samples along the growth zone and for the confirmation experiment we only used two cm pieces. Therefore, it looks like the supplementary figure represents a subset of the actual figure, but in reality the two pictures depict independent datasets. We added a sentence to clarify this in the manuscript.

Line 307, change statue to stature

The typo was corrected.

Line 339, insert comma after may

The comma was inserted.

Line 349, change 'in 5' UTR' to 'upstream'

We changed 'in the 5'UTR' to 'upstream'.

Line 350, add 'and was ligated into' after 'GA2ox)'

The text was added.

Lines 377 & 388, I am still not convinced that the stated light intensities are correct. For example full sunlight on a cloudless, clear day at high noon in the midwestern US is about 2000 $\mu\text{mol}/\text{m}^2/\text{s}$ PAR

We checked the light intensities and they are around 180-200 $\mu\text{mol}/\text{m}^2/\text{s}$ PAR instead of 170 $\text{mmol}/\text{m}^2/\text{s}$ PAR. The values were adjusted in the manuscript.

Line 418, change 'replicates' to 'replicate'

The text was adjusted accordingly.

Supplementary Table 6, column 4 the name is missing a P

The Table was adjusted accordingly.

Supplementary Figure 10, what is the difference between the two images in (c)?

The two images in what is now called supplementary figure 11C are exactly the same, just the transmitted light channel is added in the second image to illustrate that the mRFP signal is present in the phloem around the xylem because the xylem cells are only visible in the transmitted light channel. This information is added to the legend.

Reviewer #2 (Remarks to the Author):

Thank you for addressing my comments regarding the manuscript. I believe the revised version is greatly improved in quality. I think the greatest improvements are in the simplification of the presented data and expansion of the material and methods section.

I do agree with the authors that the work presented is interesting and provides novel insights. Most importantly, it provides one more valid example of how biotechnology can be used to improve crop performance. This work is definitely relevant and deserves an audience.

I appreciate the limitations inherent to field trials and propagation of materials described by the authors, but I still think the manuscript lacks the consistency in the data and novelty required for publication in **Nature communications**.

We thank the reviewer for the constructive suggestions to improve the manuscript.